# GENERALIZED RECTIFIER WAVELET COVARIANCE MODELS FOR TEXTURE SYNTHESIS

**Antoine Brochard**
ENS, PSL University, Paris, France
`antoine.brochard@ens.fr`

**Sixin Zhang**
Université de Toulouse, INP, IRIT, Toulouse, France
`sixin.zhang@irit.fr`

**Stéphane Mallat**
Collège de France, Paris, France
Flatiron Institute, New York, USA

## ABSTRACT

State-of-the-art maximum entropy models for texture synthesis are built from statistics relying on image representations defined by convolutional neural networks (CNN). Such representations capture rich structures in texture images, outperforming wavelet-based representations in this regard. However, conversely to neural networks, wavelets offer meaningful representations, as they are known to detect structures at multiple scales (e.g. edges) in images. In this work, we propose a family of statistics built upon non-linear wavelet based representations, that can be viewed as a particular instance of a one-layer CNN, using a generalized rectifier non-linearity. These statistics significantly improve the visual quality of previous classical wavelet-based models, and allow one to produce syntheses of similar quality to state-of-the-art models, on both gray-scale and color textures. We further provide insights on memorization effects in these models.

## 1 INTRODUCTION

Textures ares spatially homogeneous images, consisting of similar patterns forming a coherent ensemble. In texture modeling, one of the standard approaches to synthesize textures relies on defining a maximum entropy model (Jaynes, 1957) using a single observed image (Raad et al., 2018). It consists of computing a set of prescribed statistics from the observed texture image, and then generating synthetic textures producing the same statistics as the observation. If the statistics correctly describe the structures present in the observation, then any new image with the same statistics should appear similar to the observation. A major challenge of such methods resides in finding a suitable set of statistics, that can generate both high-quality and diverse synthetic samples. This problem is fundamental as it is at the heart of many texture related problems. For example, in patch re-arrangement methods for texture modeling, these statistics are used to compute high-level similarities of image patches (Li & Wand, 2016; Raad et al., 2018). Such models are also used for visual perception (Freeman & Simoncelli, 2011; Wallis et al., 2019; Vacher et al., 2020), style transfer (Gatys et al., 2016; Deza et al., 2019) and image inpainting (Laube et al., 2018).

A key question along this line of research is to find *what it takes to generate natural textures.* This problem was originally posed in Julesz (1962), in which the author looks for a statistical characterization of textures. In the classical work of Portilla & Simoncelli (2000) (noted PS in this work), the authors presented a model whose statistics are built on the wavelet transform of an input texture image. These statistics were carefully chosen, by showing that each of them captured a specific aspect of the structure of the image. This model produces satisfying results for a wide range of textures, but fails to reproduce complex geometric structures present in some natural texture images. Figure 1 presents a typical example composed of radishes, and synthetic images from three state-of-the-art models developed over the last few decades. To address this problem, the work of Gatys et al. (2015) proposes to use statistics built on the correlations between the feature maps of a deep CNN, pre-trained on the ImageNet classification problem (Deng et al., 2009; Simonyan & Zisserman, 2014). While this model produces visually appealing images, these statistics are hard to

interpret. The work of Ustyuzhaninov et al. (2017) made a significant simplification of such statistics, by using the feature maps of a one-layer rectifier CNN with random filters (without learning). A crucial aspect of this simplification relies on using multi-scale filters, which are naturally connected to the wavelet transform. In this paper, we propose a wavelet-based model, more interpretable than CNN-based models (with learned or random filters), to synthesize textures with complex geometric structures. It allows to bridge the gap between the classical work of Portilla & Simoncelli (2000), and state-of-the-art models.

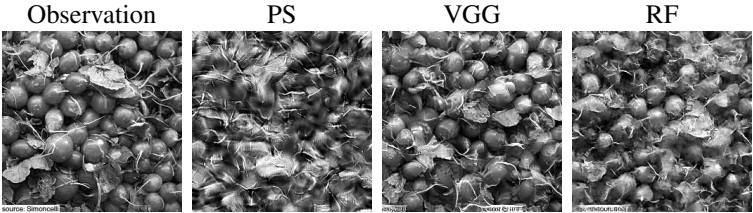

Figure 1: Example of syntheses from three texture models in chronological order. From left to right: the observed texture in gray-scale, synthesis from PS (Portilla & Simoncelli, 2000), from VGG (Gatys et al., 2015), and from RF[1](Ustyuzhaninov et al., 2017).

This model is built on the recent development of the phase harmonics for image representations and non-Gaussian stationary process modeling (Mallat et al., 2020; Zhang & Mallat, 2021). The phase harmonics are non-linear transformations that adjust the phase of a complex number. In Portilla & Simoncelli (2000); Zhang & Mallat (2021), the authors illustrate that the phase dependencies between wavelet coefficients across scales contain important information about the geometric structures in textures and turbulent flows, and that they can be captured by applying the phase harmonics to complex wavelet coefficients. Remarkably, Mallat et al. (2020) show that the phase harmonics admit a dual representation, closely related to the rectifier non-linearity in CNNs. Our main contributions are:

- We develop a family of texture models based on the wavelet transform and a generalized rectifier non-linearity, that significantly improves the visual quality of the classical wavelet-based model of Portilla & Simoncelli (2000) on a wide range of textures. It relies on introducing spatial shift statistics across scales to capture geometric structures in textures.

- By changing the number of statistics in our models, we show explicitly the trade-off on the quality and diversity of the synthesis. When there are too many statistics, our model tends to memorize image patches. We further investigate such memorization effects on non-stationary images and find that it sometimes relies on what statistics are chosen, rather than on how many.

- Through the modeling of geometric structures in gray-scale textures, our model indicates the possibility of reducing significantly the number of statistics in the works of Gatys et al. (2015) and Ustyuzhaninov et al. (2017), to achieve a similar visual quality.

The rest of the paper is organized as follows: Section 2 reviews the framework of microcanonical maximum-entropy models, build upon a general family of covariance statistics. We then present our model for both gray-scale and color textures in Section 3. Section 4 shows synthesis results of our model, compared with state-of-the-art models. Finally, in Section 5, we discuss possible improvements of our model[2].

**Notations**   Throughout the paper, $N$ denotes a positive integer. A gray-scale image $x$ is an element of $\mathbb{R}^{N \times N}$, i.e. $x = x(u)$, $u \in \Omega_N$, with $\Omega_N := \{0, \cdots, N-1\}^2$. A color image $x = \{x^c\}_{c=1,2,3}$ is an element of $\mathbb{R}^{3 \times N \times N}$, or equivalently, each $x^c \in \mathbb{R}^{N \times N}$. We shall denote $\bar{x}$ the observed texture (observation), which is assumed to be a realisation a random vector $X$. For any complex number $z \in \mathbb{C}$, $z^*$ is the complex conjugate of $z$, $\text{Real}(z)$ its real part, $|z|$ its modulus, and $\varphi(z)$ its phase.

---

[1]As the model from Ustyuzhaninov et al. (2017) uses on random filters, we shall use the abbreviation RF.

[2]All calculations can be reproduced by a Python software available at `https://github.com/abrochar/wavelet-texture-synthesis`.

## 2 MICROCANONICAL COVARIANCE MODELS

We briefly review the standard framework of micro-canonical maximum-entropy models for textures. To reliably estimate the statistics in these models, we assume that a texture is a realization of a stationary and ergodic process $X$ (restricted to $\Omega_N$). We then review a special family of statistics that are used in the state-of-the-art texture models (mentioned in Figure 1), based on covariance statistics of an image representation.

### 2.1 FRAMEWORK

Given a observation texture $\bar{x}$, we aim at generating new texture images, similar but different from $\bar{x}$. To that end, a classical method is to define a set of statistics $C\bar{x}$, computed on the observation, and try to sample from the *microcanonical set*

$$\{x : \|Cx - C\bar{x}\| \leq \epsilon\},$$

where $\| \cdot \|$ denotes the $L^2$ norm.

Under the stationary and ergodic assumption of $X$, one can construct $Cx$ as a statistical estimator of $\mathbb{E}(CX)$, from a complex-valued representation $Rx$.[3] The set of covariance statistics $Cx$ of a model can then be constructed by computing an averaging over the spatial variable $u$, i.e.

$$Cx(\gamma, \gamma', \tau) := \frac{1}{|\Omega_N|} \sum_{u \in \Omega_N} Rx(\gamma, u) Rx(\gamma', u - \tau)^*, \tag{1}$$

for $(\gamma, \gamma', \tau) \in \Upsilon \subseteq \Gamma \times \Gamma \times \Omega_N$. The statistics $Cx(\gamma, \gamma', \tau)$ can be interpreted as estimating the covariance (resp. correlations) between $RX(\gamma, u)$ and $RX(\gamma', u-\tau)$ for zero-mean $RX$ (resp. non-zero mean $RX$). The ergodicity assumption ensures that when $N$ is large enough, the approximation $C\bar{x} \simeq \mathbb{E}(CX)$ over $\Upsilon$ should hold with high probability. Under these conditions, it makes sense to sample the microcanonical set in order to generate new texture samples.

This framework encompasses a wide range of state-of-the-art texture models[4]. In particular, the PS model takes inspiration from the human early visual system to define a multi-scale representation based on the *wavelet transform* of the image (Heeger & Bergen, 1995). We next review a family of covariance model which generalizes the statistics in the PS model. We write $C^{\mathrm{M}}$ the statistics for a specific model M that uses the representation $R^{\mathrm{M}}$.

### 2.2 WAVELET PHASE HARMONIC COVARIANCE MODELS

We review a family of microcanonical covariance models defined by a representation built upon the wavelet transform and phase harmonics. It defines a class of covariance statistics that capture dependencies between wavelet coefficients across scales.

#### 2.2.1 WAVELET TRANSFORM

The wavelet transform is a powerful tool in image processing to analyze signal structures, by defining a sparse representation (Mallat, 2001). For texture modeling, we consider oriented wavelets to model geometric structures in images at multiple scales. They include the Morlet wavelets and steerable wavelets, proposed in Goupillaud et al. (1984); Simoncelli & Freeman (1995); Unser & Chenouard (2013). In particular, the Simoncelli steerable wavelets have been used to model a diverse variety of textures in Portilla & Simoncelli (2000).

Oriented wavelets are defined by the dilation and rotation of a complex function $\psi : \mathbb{R}^2 \mapsto \mathbb{C}$ on a plane. Let $r_\theta$ denote the rotation by angle $\theta$ in $\mathbb{R}^2$. They are derived from $\psi$ with dilations by factors $2^j$, for $j \in \{0, 1, \cdots, J-1\}$, and rotations $r_\theta$ over angles $\theta = \ell\pi/L$ for $0 \leq \ell < L$, where $L$ is the number of angles in $[0, \pi)$. The wavelet at scale $j$ and angle $\theta$ is defined by

$$\psi_{j,\theta}(u) = 2^{-2j}\psi(2^{-j}r_\theta u), \quad u \in \mathbb{R}^2.$$

---

[3]The complex-valued representation $Rx(\gamma, u) \in \mathbb{C}$ is a function of $(\gamma, u)$ in an index set $\Gamma \times \Omega_N$.
[4]e.g. Portilla & Simoncelli (2000); Gatys et al. (2015); Ustyuzhaninov et al. (2017); Zhang & Mallat (2021)

Scales equal or larger than $J$ are carried by a low-pass filter $\phi_J$.

The wavelet transform of an image $x \in \mathbb{R}^{N \times N}$ is a family of functions obtained by the convolution of $x$ with discrete wavelets.[5] Let $\Lambda := \{0, \cdots, J-1\} \times \frac{\pi}{L}\{0, \cdots, L-1\}$ be an index set. The wavelet coefficients are

$$x \star \psi_{j,\theta}(u) = \sum_{v \in \Omega_N} x(u-v)\psi_{j,\theta}(v), \quad u \in \Omega_N, \quad (j,\theta) \in \Lambda. \tag{2}$$

The low-pass coefficients $x \star \phi_J$ are defined similarly.

### 2.2.2 WAVELET PHASE HARMONICS AND THE PS MODEL

To model natural textures, it has been shown (Portilla & Simoncelli, 2000; Zhang & Mallat, 2021) that it is crucial to capture statistical dependencies between wavelet coefficients across scales. This can be achieved by using a wavelet phase harmonic representation, which is defined by the composition of a linear wavelet transform of $x$, and a non-linear phase harmonic transform.

In Mallat et al. (2020), the authors introduce the phase harmonics to adjust the phase of a complex number $z \in \mathbb{C}$. More precisely, the phase harmonics $\{[z]^k\}_{k \in \mathbb{Z}}$ of a complex number $z \in \mathbb{C}$ are defined by multiplying its phase $\varphi(z)$ of $z$ by integers $k$, while keeping the modulus constant, i.e. $\forall k \in \mathbb{Z}, [z]^k := |z|e^{ik\varphi(z)}$. The wavelet phase harmonic representation (WPH) is then defined by

$$R^{\mathrm{WPH}}x(\gamma, u) = [x \star \psi_{j,\theta}(u)]^k - \mu_\gamma, \quad \gamma = (j,\theta,k) \in \Gamma = \Lambda \times \mathbb{Z}, \tag{3}$$

where $\mu_\gamma$ is defined as the spatial average of $[\bar{x} \star \psi_{j,\theta}]^k$.

It is shown in Zhang & Mallat (2021) that the PS model can be regarded as a low-order wavelet phase harmonics covariance model, which considers only a restricted number of pairs $(k, k')$ (see Appendix A for more details). In the next section, we shall use a dual representation of the phase harmonic operator to define a covariance model to capture high-order phase harmonics.

## 3 GENERALIZED RECTIFIER WAVELET COVARIANCE MODEL

In the previous section, we presented a class of models, built from the wavelet phase harmonic representation. A dual representation of the phase harmonic operator $[\cdot]^k$ can be defined via a generalized rectified linear unit, that we review in Section 3.1. We then discuss in Section 3.2 how to define an appropriate index set of $\Gamma$ for gray-scale textures. Section 3.3 extends the model to color textures.

### 3.1 FROM PHASE HARMONICS TO THE GENERALIZED RECTIFIER

The generalized rectified linear unit of a complex number $z$, with a phase shifted by $\alpha \in [0, 2\pi]$, is defined by

$$\rho_\alpha(z) = \rho(\mathrm{Real}(e^{i\alpha}z)), \tag{4}$$

where $\rho$ is a rectified linear unit, i.e. for any $t \in \mathbb{R}$, $\rho(t) := \max(0, t)$. In Mallat et al. (2020), it is shown that applying a Fourier transform on $\rho_\alpha(z)$ along the variable $\alpha$ results in the phase harmonics of $z$ (up to some normalization constant). This suggests an alternative model, defined by coefficients of the form

$$R^{\mathrm{ALPHA}}x(\gamma, u) = \rho_\alpha(x \star \psi_{j,\theta}(u)) - \mu_\gamma, \quad \gamma = (j,\theta,\alpha), \tag{5}$$

for $\gamma \in \Gamma = \Lambda \times [0, 2\pi]$, and $\mu_\gamma$ is defined as the spatial average of $\rho_\alpha(\bar{x} \star \psi_{j,\theta}(u))$ over $u \in \Omega_N$.

**Relation with high-order phase harmonics** Based on the duality between the phase harmonics $k \in \mathbb{Z}$ and the phase shift variable $\alpha \in [0, 2\pi]$, we now present the relation between $C^{\mathrm{ALPHA}}$ and the high-order phase harmonics in $C^{\mathrm{WPH}}$, first proved in Mallat et al. (2020).

**Proposition 1** *There exists a complex-valued sequence $\{c_k\}_{k \in \mathbb{Z}}$ such that for all $(j,\theta,\alpha) \in \Gamma$, $(j',\theta',\alpha') \in \Gamma$, and all $\tau \in \Omega_N$,*

$$C^{\mathrm{ALPHA}}x((j,\theta,\alpha),(j',\theta',\alpha'),\tau) = \sum_{(k,k') \in \mathbb{Z}^2} c_k c_{k'}^* C^{\mathrm{WPH}}x((j,\theta,k),(j',\theta',k'),\tau)e^{i(k\alpha - k'\alpha')}.$$

---

[5]The continuous wavelets are discretized with periodic boundary conditions on the spatial grid $\Omega_N$.

The proof is given in Appendix B. We remark that the sequence $\{c_k\}_{k\in\mathbb{Z}}$ is uniquely determined by the rectifier non-linearity $\rho$, and they are non-zero if $k$ is even (Mallat et al., 2020). This result shows that for a suitable choice of $(\alpha, \alpha')$, the covariance statistics $C^{\text{ALPHA}}x$ can implicitly capture $C^{\text{WPH}}x$ with a wide range of $k$ and $k'$.

**Relation with second order statistics**  Using a simple decomposition of wavelet coefficients into their positive, negative, real and imaginary parts, we can further show that the covariance statistics $C^{\text{ALPHA}}x$ capture the classical second order statistics of wavelet coefficients, also used in the PS model (with phase harmonic coefficients $k = k' = 1$).

**Proposition 2** *Let* $I = \{0, \frac{\pi}{2}, \pi, \frac{3\pi}{2}\}$. *There exists a finite complex-valued sequence* $\{w_{\alpha,\alpha'}\}_{(\alpha,\alpha')\in I^2}$ *such that for all* $(j,\theta) \in \Lambda$, *and all* $\tau \in \Omega_N$,

$$\sum_{(\alpha,\alpha')\in I^2} w_{\alpha,\alpha'} C^{\text{ALPHA}}x((j,\theta,\alpha),(j',\theta',\alpha'),\tau) = \sum_{u\in\Omega_N} \left(x \star \psi_{j,\theta}(u)\right)\left(x \star \psi_{j',\theta'}(u-\tau)\right)^*. \quad (6)$$

The proof is given in Appendix C. This shows that using only four $\alpha$ uniformly chosen between $[0, 2\pi]$ is sufficient to capture second order statistics. Because the wavelet transform is an invertible linear operator (on its range space), computing the r.h.s of eq. (6) for all $(j, \theta, \tau)$, as well as the low-pass coefficients carried out by $\Phi_J$, is equivalent to computing the correlation matrix of $x$.

**Relation with the RF model**  Setting aside the subtraction by the spatial mean $\mu_\gamma$, the RF model can be viewed as a particular case of models defined by eq. (5). Indeed, the statistics of the RF model take the form of eq. (1), with

$$R^{\text{RF}}x(f, u) = \rho(x \star \psi_f(u)),$$

where $\{\psi_f\}$ being a family of multi-scale random filters. By writing $\rho_\alpha(x \star \psi_{j,\theta}(u-\tau)) = \rho(x \star \text{Real}(\psi_{j,\theta}^\tau e^{i\alpha})(u))$, with $\psi_{j,\theta}^\tau$ denoting the translation of $\psi_{j,\theta}$ by $\tau$, we see that the models are similar, the difference being that our models use wavelet-based filters instead of random ones.

### 3.2 DEFINING AN APPROPRIATE $\Upsilon$

The choice of the covariance set $\Upsilon$ is of central importance in the definition of the model. Intuitively, a too small set of indices will induce a model that could miss important structural information about the texture that we want to synthesize. Conversely, if $\Upsilon$ contains too many indices, the syntheses can have good visual quality, but the statistics of the model may have a large variance, leading to the memorization of some patterns of the observation. There is a trade-off between these two aspects: one must capture enough information to get syntheses of good visual quality, but not much, so as not to reproduce parts of the original image. To illustrate this point, we shall study the model ALPHA defined with three different sets $\Upsilon$ : A smaller model $\text{ALPHA}_S$ with a limited amount of elements in $\Upsilon$, an intermediate model $\text{ALPHA}_I$, and a larger model $\text{ALPHA}_L$.

To precisely define these models, let us note $\mathfrak{J} := \{0, \cdots, J-1\}$, $\Theta := \frac{\pi}{L}\{0, \cdots, L-1\}$, and $\mathcal{A}_A = \frac{2\pi}{A}\{0, \cdots, A-1\}$. Let us also define the set $\mathfrak{T} := \{0\} \cup \{2^j(\cos(\theta), \sin(\theta))\}_{0\le j<J, \theta\in\frac{\pi}{L}\{0,\cdots,2L-1\}}$, from which the spatial shift shall be selected. Table 1 summarizes the conditions that all parameters have to satisfy to be contained in these sets. Additionally, these models include large scale information through the covariance of a low-pass filter, i.e. the spatial average of $x \star \phi_J(\cdot)\overline{x \star \phi_J(\cdot - \tau)}$, for $\tau \in \mathfrak{T}$. To count the size of $\Upsilon$ without redundancies, Appendix A.3 provides an upper bound on the non-redundant statistics in our models. This upper bound is used to count the number of statistics in our models. To keep this number from being too large, instead of taking all shifts in a square box, such as in Portilla & Simoncelli (2000), we choose to select only shifts of dyadic moduli, and with the same orientations as the wavelets.

$\text{ALPHA}_S$ **vs.** $\text{ALPHA}_I$  The small model $\text{ALPHA}_S$ is inspired from the PS model, as it only takes into account of the interactions between nearby scales (i.e. $|j' - j| \le 1$), and the spatial shift correlations are only considered for $(j,\theta) = (j',\theta')$. There are two notable differences in the statistics included in the $\text{ALPHA}_S$ and $\text{ALPHA}_I$ models. The first one is the range of scales being correlated. It has been shown in Zhang & Mallat (2021) that constraining correlation between a wider

Table 1: List of indices in $\Upsilon$ for different ALPHA models.

| Model | Scales | Angles | Phase shift | Spatial shift | Size of $\Upsilon$ |
|---|---|---|---|---|---|
| $\text{ALPHA}_\text{S}$ | $\substack{(j,j')\in\mathfrak{J}^2 \\ |j'-j|\le 1}$ | $(\theta,\theta')\in\Theta^2$ | $(\alpha,\alpha')\in\mathcal{A}_4\times\mathcal{A}_1$ | $\substack{\tau\in\mathfrak{T}\text{ if }(j,\theta)=(j',\theta') \\ \tau=0\text{ otherwise.}}$ | $(J|\Theta|^2 + J|\Theta||\mathfrak{T}|)|\mathcal{A}_4|$ |
| $\text{ALPHA}_\text{I}$ | $(j,j')\in\mathfrak{J}^2$ | $(\theta,\theta')\in\Theta^2$ | $(\alpha,\alpha')\in\mathcal{A}_4\times\mathcal{A}_1$ | $\tau\in\mathfrak{T}$ | $J^2|\Theta|^2|\mathcal{A}_4||\mathfrak{T}|$ |
| $\text{ALPHA}_\text{L}$ | $(j,j')\in\mathfrak{J}^2$ | $(\theta,\theta')\in\Theta^2$ | $(\alpha,\alpha')\in\mathcal{A}_4^2$ | $\tau\in\mathfrak{T}$ | $J^2|\Theta|^2|\mathcal{A}_4|^2|\mathfrak{T}|$ |

range of scales induces a better model for non-Gaussian stationary processes, and a better estimation of cosmological parameters from observed data (Allys et al., 2020). The second difference, which has a significant impact on the number of statistics (it increases the model size by a factor $\sim$10), is the number of spatial shifts in the correlations. In the $\text{ALPHA}_\text{I}$ model, spatially shifted correlations are computed for all pairs of coefficient $(\gamma, \gamma')$. For both stationary textures and non-stationary images in gray-scale, shape and contours of salient structures and objects are better reproduced with $\text{ALPHA}_\text{I}$, as illustrated in Figure 2. More examples are given in Appendix D.

$\text{ALPHA}_\text{I}$ **vs.** $\text{ALPHA}_\text{L}$    As we observe in Figure 2, the $\text{ALPHA}_\text{I}$ model, containing 4 times less coefficients than the $\text{ALPHA}_\text{L}$, suffers less from memorization effects, while still capturing most of the geometric information in the images. This small loss of information can be partially explained by the frequency transposition property of the phase harmonics operator (Mallat et al., 2020), for compactly supported wavelets in the frequency domain, as detailed in Appendix E. In order to avoid this memorization effect, we shall, in the rest of the paper, consider only the intermediate model.

### 3.3 Modelling color interactions

In order to generate color textures, the covariance model $\text{ALPHA}_\text{I}$ defined in Section 3.2 could be directly applied to each R, G and B color channel independently. However, it would not take into account the color coherence in the structures of the observation.

To capture color interactions in the observation image, we shall impose the covariance between the coefficients of eq. (5) for all indices in $\Upsilon$ and all color channels. More precisely, let $x = \{x^c\}_{c=1,2,3}$ be a color image, with the parameter $c$ representing the color channel. The $\text{ALPHA}_\text{C}$ color model is defined by correlations between coefficients of the form:

$$R^{\text{ALPHA}_\text{C}}x(\gamma, u) = \rho_\alpha(x^c \star \psi_{j,\theta})(u) - \mu_\gamma, \quad \gamma = (j, \theta, \alpha, c). \tag{7}$$

The set of indices is defined as $\Upsilon^{\text{ALPHA}_\text{C}} := \{(\gamma, \gamma', \tau) : ((j,\theta,\alpha),(j',\theta',\alpha'),\tau) \in \Upsilon^{\text{ALPHA}_\text{I}}, (c,c') \in \{1,2,3\}^2\}$.

**Reduced** $\text{ALPHA}_\text{C}$    The model $\text{ALPHA}_\text{C}$ has a large size as it computes correlations between all coefficients for all color channels. This size can be significantly reduced by computing spatially shifted coefficients only for the same color channels (to capture their geometries). This reduced model contains three times less coefficients ($\sim$113k), without significant reduction of the visual quality of syntheses, as detailed in Appendix F.

## 4 Numerical results

In this section, we compare our intermediate model to the state-of-art models (PS, RF and VGG) on both gray-scale and color textures. We first specify the experimental setup. We then present the synthesis results of various examples, and discuss their quality through visual inspection. A quantitative evaluation of the quality of the syntheses, based on the synthesis loss of the VGG model, and proposed in Ustyuzhaninov et al. (2017), is discussed in Appendix G.

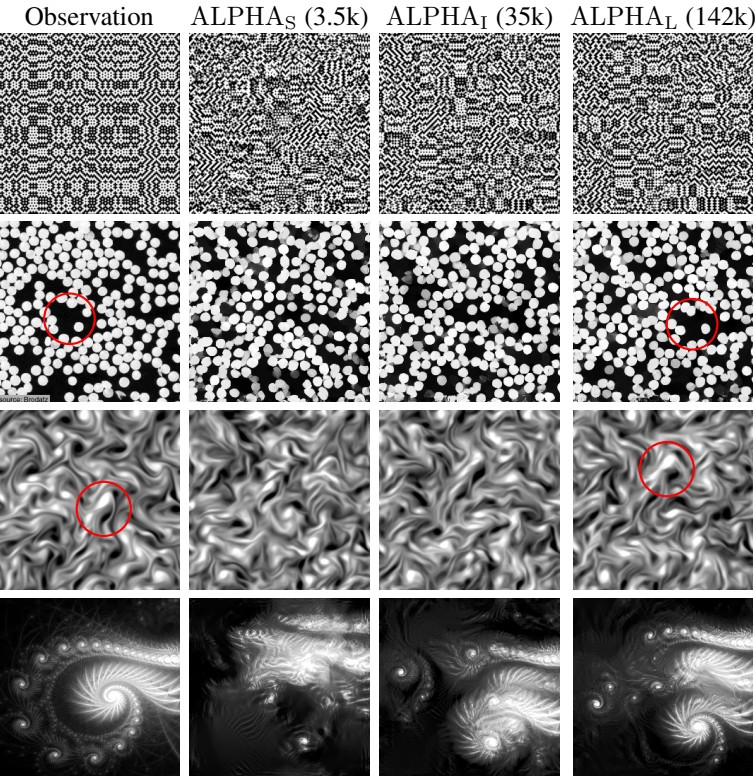

Figure 2: Examples of syntheses from the ALPHA models defined in Table 1. Visually similar image patches in textures are highlighted by red circles. The number of statistics is given in brackets next to each model name. From top to bottom: a Julesz counterexample, a stationary texture image, a stationary turbulent field and a non-stationary image. We see that the model $ALPHA_I$ achieves a balance between the visual quality and diversity on these examples.

## 4.1 EXPERIMENTAL SETUP

For our experiments, we choose gray-scale and color textures with various visual structures.[6] In the gray-scale examples, we also include a stationary turbulent field (vorticity), which is simulated from Navier-Stokes equations in two dimensions (Schneider et al., 2006). These examples all contain complex geometric structures that are hard to model by the classical PS model.

The texture images presented in this work have a size of $N = 256$, giving a number of pixels of ∼65k. For all the ALPHA models, we use Morlet wavelets with a maximal scale $J = 5$ and number of orientations $L = 4$. This choice differs from the PS model, which uses Simoncelli steerable wavelets. In Appendix H, we discuss the impact of these wavelets on our model. To draw the samples, we follow gradient-based sampling algorithms, suitable in high-dimensions (Bruna & Mallat, 2018), to minimize the objective function $\|Cx - C\bar{x}\|^2$, starting from an initial sample from a normal distribution. Similarly to Gatys et al. (2015); Ustyuzhaninov et al. (2017), we use the L-BFGS algorithm (Nocedal, 1980) for the optimization of the objective. As in the VGG model (Gatys et al., 2015), we further apply a histogram matching procedure as post-processing after optimization. The details of the PS, RF and VGG models, as well as detailed specifications of our models, are given in Appendix A. More synthesis examples can be found in Appendix I.

## 4.2 RESULTS

In Figure 3, we present examples of syntheses from the $ALPHA_I$ (or $ALPHA_C$), PS, RF and VGG models, for both gray-scale and color textures, as well as for non-stationary images. We observe that

---

[6]The source for the presented textures are given in Appendix A.

our model $\text{ALPHA}_\text{I}$ produces texture syntheses of similar visual quality to the RF and VGG models. It also significantly outperforms the PS model in terms of the visual quality, without introducing visible memorization effects. As the model PS uses the statistics closer to $\text{ALPHA}_\text{S}$ compared to $\text{ALPHA}_\text{I}$, the performance of PS is somehow expected.

Note that for the tiles example (the fifth row) in Figure 3, the VGG model produces less convincing textures, because the long-range correlations present in the image (aligned tiles) are not reproduced. To remedy this issue, it has been proposed in Berger & Memisevic (2017) to add spatial shifts to the correlations of the network feature maps. These shift statistics are similar to the parameter $\tau$ in our model. We also observe that, in the case of the sixth row example (flowers), all models fail to reproduce complex structures at object-level. Possible improvements of such models is further discussed in Section 5.

For non-stationary images, we find that certain image patches can be more or less memorized by the RF, VGG and $\text{ALPHA}_\text{I}$ models, as illustrated in the seventh row example. Understanding such memorization effect of non-stationary images is a subtle topic, as we find that in some binary images ($\bar{x}(u) \in \{0, 1\}$), only the PS model can reproduce the observation, even though it has a much smaller number of statistics (the last row example). We find that this is related to the spatial correlation statistics in PS (non-zero $\tau$). By removing this constraint, $\bar{x}$ is no longer always reproduced.[7] The non-stationary nature also appear in some logo near the boundary of some textures (e.g. bottom left in the observation of the first and fourth rows). Although this logo is reproduced by RF, VGG, $\text{ALPHA}_\text{I}$ and $\text{ALPHA}_\text{C}$, it is a very local phenomenon, as we do not find visible copies of the textures when there is no logo, and it is likely related to the way one addresses the boundary effect (see more in Appendix A.5).

## 5 DISCUSSION

In this work, we presented a new wavelet-based model for natural textures. This model incorporates a wide range of statistics, by computing covariances between rectified wavelet coefficients, at different scales, phases and positions. We showed that this model is able to capture and reproduce complex geometric structures, present in natural textures or physical processes, producing syntheses of similar quality to state-of-the-art models that use CNN-based representations.

Being defined with a wavelet family instead of multi-scale random filters, the proposed model uses less statistics than the RF model. For the gray-scale textures, our model has about 15 times less statistics, as it focuses on capturing the geometric structures present in images. Although our color model has a slightly larger number of statistics than VGG, the reduced color model, presented in Section 3.3 is three times smaller than $\text{ALPHA}_\text{C}$, while achieving similar visual quality. It shows the potential to further reduce the size of the color model. In the PS model, a PCA on the color channels is performed (Vacher & Briand, 2021). The same idea could also be applied to our model.

Furthermore, there are examples where all the models may all fail to produce some geometric structures at object-level, as illustrated in Figure 3 (sixth row). In this situation, we still need to find more informative statistics. One may for example consider to incorporate a second layer of wavelet transform as in the wavelet scattering transform (Leonarduzzi et al., 2019; He & Hirn, 2021). Another line of research is to introduce other kinds of losses (such as to encourage image smoothness) in order to improve the VGG model (Liu et al., 2016; Sendik & Cohen-Or, 2017). These losses are complementary, and could thus also be added to our models. Integrating these models with learning-based approaches is another promising direction (Zhou et al., 2018; Zhu et al., 1997; Xie et al., 2016; 2018).

Finding a minimal set of statistics to define a texture model remains important because a large number of statistics can result in a high variance of the estimators, and the associated model may suffer from memorization effects. This is a problem because the aim of the model is to approximate the underlying distribution of the observation, and therefore produce diverse textures. In this regard, the mere visual evaluation of the synthetic textures can fail to take this aspect of the model into account. Defining a quantitative evaluation of quality *and* diversity, coherent with visual perception, remains an open problem (Ustyuzhaninov et al., 2017; Yu et al., 2019).

---

[7]Set the parameter $\Delta = 0$ in the PS model. See Appendix A for more details. This simple example suggests that sometimes it is very important to choose the right statistics to capture specific geometric structures.

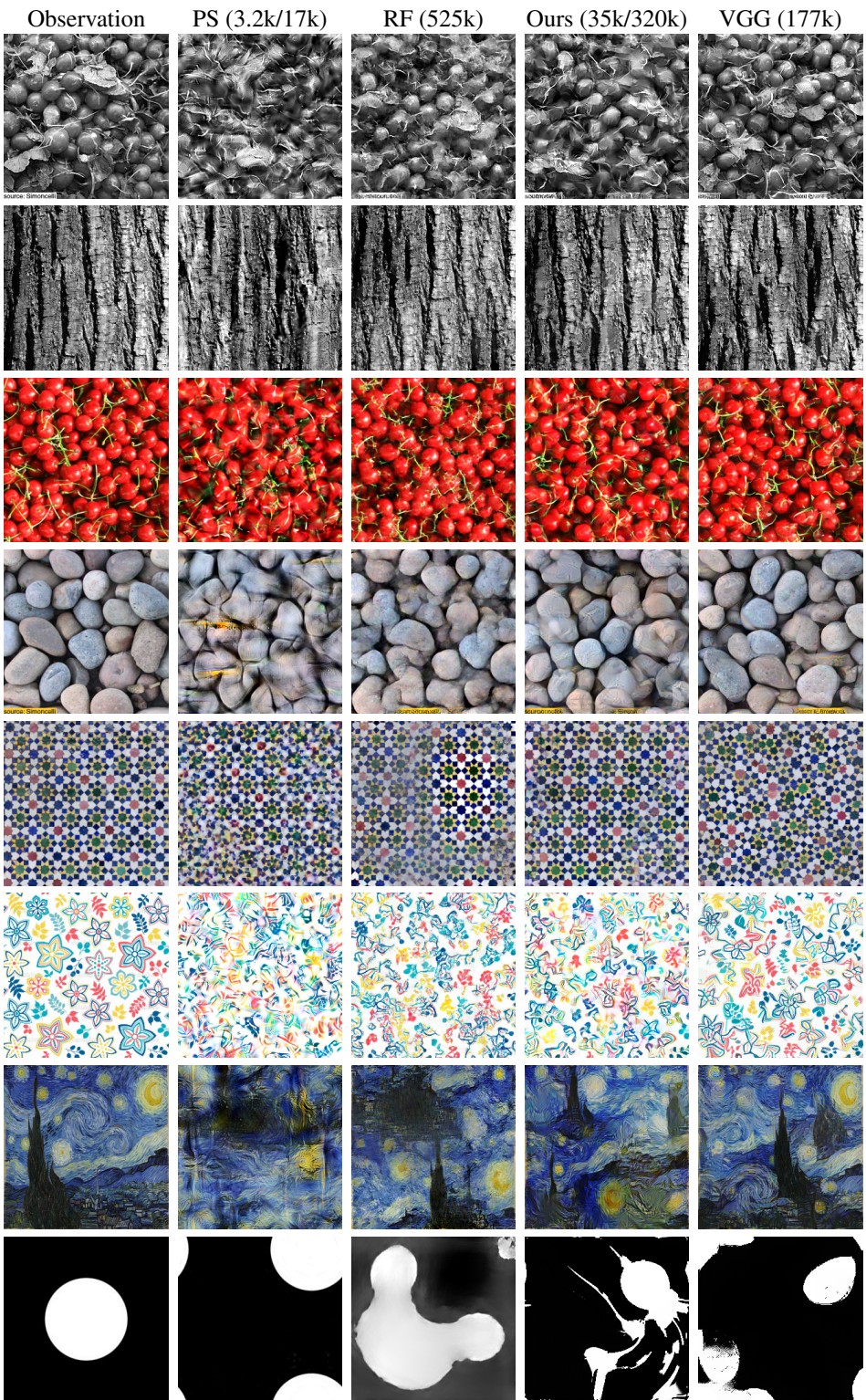

Figure 3: Visual comparison between the gray-scale and color models. 'Ours' denotes the $\text{ALPHA}_\text{I}$ model for gray-scale images and the $\text{ALPHA}_\text{C}$ model for color images.

**Acknowledgments**  We thank all the reviewers for their insightful feedback. This work was partially supported by a grant from the PRAIRIE 3IA Institute of the French ANR-19-P3IA- 0001 pro-

gram. Sixin Zhang acknowledges the support provided by 3IA Artificial and Natural Intelligence Toulouse Institute, French "Investing for the Future - PIA3" program under the Grant agreement ANR-19-PI3A-0004, and by Toulouse INP ETI and AAP unique CNRS 2022 under the Project LRGMD.

**Reproducibility Statement**    In order to reproduce the experiments presented in this work, the main parameters to define the model can be found in Section 4.1. More details are given in Appendix A. Additionally, the code is made publicly available.

**Ethics Statement**    The authors acknowledge that no potential conflicts of interest, discrimination, bias, fairness concerns or research integrity issues have been raised during the completion of this work.

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

## A  MODEL AND ALGORITHMIC SPECIFICATION

We provide additional information needed to reproduce the numerical results of the models (for both gray-scale and color textures) considered in this paper. First, we detail the models of PS, RF, VGG. We then give algorithmic parameters to obtain the synthesis images. For natural textures, we also propose a strategy to synthesize non-periodic images in our models. An image is non-periodic if a periodic extension of the image to the domain outside $\Omega_N$ create discontinuities at the contour of $\Omega_N$.

**Sources of textures**  Our natural texture examples were obtained from the following three sources: CNS NYU[8], Textures.com[9], Describable Textures Dataset model[10] and the Github page of Berger & Memisevic (2017) [11].

### A.1  MODEL PARAMETERS

We specify the model parameters to synthesize both gray-scale and color textures. We also discuss how to extend the RF and the VGG model, originally designed for color textures, to model gray-scale textures.

- PS: For both gray-scale and color models, we set the number of scales $J = 5$, and the number of orientations $L = 8$ for the Simoncelli steerable wavelets. The spatial shift $\tau = (\tau_1, \tau_2) \in \Omega_N$ is chosen to be in the range of $\max(\tau_1, \tau_2) \leq \Delta = 4$. Note that this is different to the model parameters reported for the default PS model (which is $J = 4, L = 4, \Delta = 3$) as we find that it results in a larger set of statistics and better visual quality. The synthesis results of this model can be reproduced by a Matlab software.[12]

- RF: For gray-scale textures, we consider $J \times L$ random convolutional filters $(\psi_f)_{1 \leq f \leq JL}$. Let $f = (j, \ell)$, the index $j$ representing the scale of each filter, whose size is $(W_j, W_j)$ for $1 \leq j \leq J$. For $\Gamma = \{f = (j, \ell) | j \leq J, \ell \leq L\}$, the representation is

$$R^{\mathrm{RF}}x(\gamma, u) = \rho(x \star \psi_{j,\ell}(u)), \quad \gamma = (j, \ell) \in \Gamma.$$

For a color image $x = \{x^c\}_{1 \leq c \leq 3}$, we use $3 \times J \times L$ random convolutional filters. The representation of $x$ is

$$R^{\mathrm{RF}}x(\gamma, u) = \rho\Big(\sum_{c=1}^3 x^c \star \psi_{c,j,\ell}(u)\Big), \quad \gamma = (j, \ell) \in \Gamma.$$

The correlations $C^{\mathrm{RF}}x$ are defined for all pairs $(\gamma, \gamma') \in \Gamma \times \Gamma$. In both the gray and color cases, it results in a correlation matrix $C^{\mathrm{RF}}$ with $J^2L^2$ statistics.

---

[8] http://www.cns.nyu.edu/~lcv/texture/
[9] https://textures.com/
[10] https://www.robots.ox.ac.uk/~vgg/data/dtd/index.html
[11] https://github.com/guillaumebrg/texture_generation
[12] https://www.cns.nyu.edu/~lcv/texture/

Following the default setting of the RF model, we set $J = 8$ and $L = 128$ for filters whose sizes are $W_1 = 3, W_2 = 5, W_3 = 7, W_4 = 11, W_5 = 15, W_6 = 23, W_7 = 37, W_8 = 55$. Each filter $\psi_{j,\ell}$ or $\psi_{c,j,\ell}$ is generated randomly according to `GlorotUniform` in the software Lasagne. [13]

- VGG. For a color image $x = \{x^c\}$, the VGG model computes a correlation matrix $C^{\mathrm{VGG}} x$ between the features maps within different layers of a pre-trained CNN network. To adapt this model to gray-scale textures, we shall add one input layer which converts a gray-scale image $y$ into a color image, by setting $x^c = y$ for each color channel $c$. This allows one to use the same $C^{\mathrm{VGG}} x$ to compute the gradient of the VGG loss with respect to $y$, and therefore to synthesise a gray-scale texture. For both gray-scale and color textures, we use only five layers 'conv1_1', 'pool1', 'pool2', 'pool3', 'pool4', as proposed in the original work.

- ALPHA. See the main text.

## A.2 Algorithmic parameters

We specify the optimization parameters used to synthesize both gray-scale and color textures.

- PS: It utilizes iterative projections onto constraint sets to generate textures. We set the number of iterations to 200.

- RF: It uses the L-BFGS procedure[14] with a memory size 20, and with a maximal number of iterations 2000. The initialization for each pixel value of a gray-image is Uniform between $[-1, 1]$. For the color image case, each RGB channel is initialized independently with a Uniform distribution between $[-1, 1]$. To address non-zero mean textures (i.e. $\mathbb{E}(X(u)) \neq 0$), the empirical mean of $\bar{x}$ is subtracted from the input $x$ to compute the representation. It is added back to the output of the optimization to produce a synthesis.

- VGG: It uses the L-BFGS procedure[15] with a memory size of 20, and with a maximal number of iterations of 2000. For both gray-scale and color images, bounds constraints are used for the optimization. The initialization of each pixel value is the standard normal distribution (zero mean, unit variance). To address non-zero mean textures (i.e. $\mathbb{E}(X(u)) \neq 0$), the VGG mean is subtracted from the input $x$ of the representation[16]. It is added back to the output after the optimization to produce a synthesis (with an additional histogram matching post-processing).

- ALPHA: For all the models in ALPHA, we use the L-BFGS optimization algorithm with restarts. Starting from the standard normal distribution, with mean and standard deviation estimated from the observation, we use the L-BFGS procedure implemented in Pytorch. It runs for 500 iterations and then it is restarted with an initialization obtained from the previous L-BFGS result. This is repeated 10 times to obtain the synthesis (with an additional histogram matching post-processing).

## A.3 Number of statistics in the ALPHA models

In this section, we detail the number of statistics in the different ALPHA models. We begin by giving the formula for each model (note that we do not include the low-pass statistics, which numbers are negligible). To (partially) avoid redundancy in the coefficients, for all models, we compute only the correlations for indices in $\Upsilon$ such that $j_2 \geq j_1$. This gives us the following formulas for the number of statistics:

- $\#(\mathrm{ALPHA}_S) = (2J - 1)|\Theta_4|^2|\mathcal{A}_4| + J|\Theta_4||\mathcal{A}_4||\mathfrak{T}|$
- $\#(\mathrm{ALPHA}_I) = \frac{1}{2}J(J + 1)|\Theta_4|^2|\mathcal{A}_4||\mathfrak{T}|$

---

[13] https://lasagne.readthedocs.io/

[14] `scipy.optimize.fmin_l_bfgs_b` in Python

[15] `scipy.optimize.minimize` in Python

[16] For the color image whose pixel value is between zero and one, the mean of BGR is 0.40760392, 0.45795686, 0.48501961. For the gray-scale, we simply take the average of the BGR mean.

- $\#(\text{ALPHA}_\text{L}) = \frac{1}{2} J(J+1)|\Theta_4|^2|\mathcal{A}_4|^2|\mathfrak{T}|$
- $\#(\text{ALPHA}_\text{C}) = \frac{9}{2} J(J+1)|\Theta_4|^2|\mathcal{A}_4||\mathfrak{T}|$

Note that, for all ALPHA models, we also compute first order statistics $\mu_\gamma$, i.e. the spatial averages of $R^{\text{ALPHA}}\bar{x}(\gamma, u)$. There are $J|\Theta_4||\mathcal{A}_4|$ statistics of this sort in every model, which is negligible with respect to the total number of second order statistics.

Note also that there are still some redundancies in these statistics, as for $j = j'$, all correlations for $\theta, \theta', \alpha, \alpha', \tau$ are counted twice. The number of such statistics is[17], for all model, superior to the number of first order statistics, which shows that our formula is in fact an upper bound for the exact number of statistics.

## A.4 NUMBER OF STATISTICS IN THE PS MODELS

The PS model can be interpreted as a particular case of the WPH covariance model as it contains the following three key categories of statistics:

• Raw coefficient correlations ($k = k' = 1$): they capture 2nd order statistics of a stationary process $X$, i.e. the correlations between $X(u)$ and $X(u')$ for $(u, u') \in \Omega_N^2$.

• Coefficient magnitude statistics ($k = k' = 0$): they capture information of $X$ beyond the 2nd order statistics. Very often, nearby scales and angles are considered in the model, such as $j' \approx j$ and $\theta' \approx \theta$.

• Cross-scale phase statistics ($k = 1, k' = 2$): they capture local phase alignments of the wavelet coefficients at nearby scales $j' = j + 1$, which are complementary to the magnitude correlations.

The major issue to count the number of statistics of this model is to avoid double counting the statistics which are the same. This is mostly due to the symmetries of the covariance matrices. We next detail how obtain the number of statistics for both the gray-scale and color model, by following the work of Portilla & Simoncelli (2000) and Vacher & Briand (2021).

As before, we assume the number of wavelet scales is $J$, the number of wavelet orientations is $L$, and the spatial shift range is within a square of size $(2\Delta+1) \times (2\Delta+1)$. Let us denote $N_a = 2\Delta+1$. We next describe in detail the number of statistics of each category counted in our paper,

### A.4.1 PS IN GRAY-SCALE

• Marginal statistics of $x$: 6. They include mean, variance, skewness, etc.

• Marginal statistics of wavelet coefficients: $2(J+1)+1$.

• Auto-correlation of wavelet coefficients (raw coefficient correlations): $(J+1)(N_a^2+1)/2$.

• Auto-correlation of magnitude of wavelet coefficients (coefficient magnitude statistics): $JL(N_a^2 + 1)/2 + JL(L-1)/2 + (J-1)L^2$.

• Mean of magnitude of wavelet coefficients: $JL + 2$. This is not counted in the paper of Portilla & Simoncelli (2000), but it used in the Matlab software.

• Cross-correlation of phase of wavelet coefficients (cross-scale phase statistics): $2(J-1)L^2 + JL^2$. The extra $JL^2$ coefficients are cross-correlation of real sub-band cousin coefficients, which are not counted in the paper, but used in the Matlab software.

To compare with the model size of the original work of Portilla & Simoncelli (2000), one can check that the sum of these number is 792 when $J = 4, L = 4, \Delta = 3$ ($N_a = 7$). If we do not count the coefficients which are only counted in the Matlab software ($JL + 2 + JL^2 = 82$), it results in 710 as reported in the original paper.

### A.4.2 PS IN COLOR

• Marginal statistics of $x$ and PCA transform of $x$: $6 \times 3 + 3 \times 4 = 30$.

---

[17]The number of such moments is of the order of $J|\Theta|^2|\mathcal{A}_4|$ for the small model, $J|\Theta|^2|\mathcal{A}_4|^2$ for the intermediate and large models, and $9J|\Theta|^2|\mathcal{A}_4|^2$ for the color model.

- Marginal statistics of wavelet coefficients: $6(J+1) + 9$.

- Auto-correlation of wavelet coefficients (raw coefficient correlations): $3(J+2)(N_a^2+1)/2$. This is called Central autoCorr of the PCA bands in the Matalb software (which includes lowband).

- Auto-correlation of magnitude of wavelet coefficients (coefficient magnitude statistics): $3JL(N_a^2+1)/2 + J(3L)(3L-1)/2 + (J-1)(3L)^2$.

- Mean of magnitude of wavelet coefficients: $3(JL+2)$.

- Cross-correlation of phase of wavelet coefficients (cross-scale phase statistics): $J(3L)^2 + (J-1)(3L)(6L)$.

### A.5 Non periodic boundaries in natural images

The convolution operation in the wavelet transform (equation 2) is performed using the Fast Fourier Transform. Additionally, recall from Section 2.1 that spatial shifts are defined with periodic boundary conditions. This implies periodicity of the input image $x$. However, natural texture images are not periodic, so one needs to adapt the computation of coefficients to take into account possible border effects. To that end, instead of averaging over all $u \in \Omega_N$ as in eq. (1), each correlation coefficient is averaged over a sub-window inside $\Omega_N$, which size depends on the scales of the coefficients being correlated. More precisely, let $\gamma = (j, \theta, \alpha)$ and $\gamma' = (j', \theta', \alpha')$. Note $j_m := \max(j, j')$. We define $\Omega_{j_m} := \{u = (u_1, u_2) \in \Omega_N : 2^{j_m} \le u_i < N - 2^{j_m}, i = 1, 2\}$. Then, for non periodic images, we compute

$$C^{\text{ALPHA}} x(\gamma, \gamma', \tau) = \frac{1}{|\Omega_{j_m}|} \sum_{u \in \Omega_N} \mathbb{1}_{\Omega_{j_m}}(u) \mathbb{1}_{\Omega_{j_m}}(u - \tau) R^{\text{ALPHA}} x(\gamma, u) R^{\text{ALPHA}} x(\gamma', u - \tau),$$

(8)

where the spatial shifts are defined periodically. Note that the spatial averages $\mu_\gamma$ and $\mu_{\gamma'}$ are also performed on $\Omega_{j_m}$.

## B Proof of Proposition 1

Using the fact that

$$\rho(\text{Real}(ze^{i\alpha})) = \rho(\text{Real}(|z|e^{i(\varphi(z)+\alpha)}))$$
$$= |z|\rho(\cos(\alpha + \varphi(z))),$$

and computing the Fourier coefficients of the $2\pi$-periodic function $\rho_\alpha(z)$ in the variable $\alpha$, we obtain

$$\mathcal{F}(\rho_\alpha(z))(k) := \frac{1}{2\pi} \int_{[0,2\pi]} \rho_\alpha(z) e^{-ik\alpha} d\alpha$$
$$= |z| \frac{1}{2\pi} \int_{[0,2\pi]} \rho(\cos(\alpha + \varphi(z))) e^{-ik\alpha} d\alpha$$
$$= |z| e^{ik\varphi(z)} c_k$$
$$= [z]^k c_k,$$

where $c_k$ is the Fourier transform of $h(.) := \rho(\cos(.))$ at the frequency $k$. The function $\alpha \mapsto \rho_\alpha(z)$ being periodic in $\alpha$, we have its decomposition in Fourier series

$$\rho_\alpha(z) = \sum_{k \in \mathbb{Z}} \mathcal{F}(\rho_\alpha(z))(k) e^{ik\alpha}$$
$$= \sum_{k \in \mathbb{Z}} c_k [z]^k e^{ik\alpha}.$$

We can then write, for any $z, z' \in \mathbb{C}$, and $\alpha, \alpha' \in [0, 2\pi]$,

$$\rho_\alpha(z) \rho_{\alpha'}(z')^* = \sum_{k, k' \in \mathbb{Z}^2} c_k c_{k'}^* [z]^k [z']^{-k'} e^{i(k\alpha - k'\alpha')}.$$

Replacing $z$ and $z'$ by any two wavelet coefficients $x \star \psi_{j,\theta}(u)$ and $x \star \psi_{j',\theta'}(u - \tau)$, we thus obtain the relation in Proposition 1.

## C    PROOF OF PROPOSITION 2

Let $z \in \mathbb{C}$, and recall from eq. (4), that $\rho_\alpha(z) = \rho(\text{Real}(ze^{i\alpha}))$. Note that we have the following relation

$$z = \rho_0(z) - \rho_\pi(z) - i(\rho_{\frac{\pi}{2}}(z) - \rho_{\frac{3\pi}{2}}(z)). \tag{9}$$

We can then write

$$
\begin{aligned}
zz'^* &= \big(\rho_0(z) - \rho_\pi(z) - i(\rho_{\frac{\pi}{2}}(z) - \rho_{\frac{3\pi}{2}}(z))\big)\big(\rho_0(z') - \rho_\pi(z') - i(\rho_{\frac{\pi}{2}}(z') - \rho_{\frac{3\pi}{2}}(z'))\big) \\
&= \sum_{\alpha,\alpha' \in I^2} w'_{\alpha,\alpha'} \rho_\alpha(z)\rho_{\alpha'}(z'),
\end{aligned}
$$

with $I = \{0, \frac{\pi}{2}, \pi, \frac{3\pi}{2}\}$. Replacing $z$ with $x \star \psi_{j,\theta}(u)$, $z'$ with $x \star \psi_{j',\theta'}(u - \tau)$,, and injecting this relation in eq. (1) gives us the desired result, with $w_{\alpha,\alpha'} = \Omega_N w'_{\alpha,\alpha'}$.

## D    SUPPLEMENTARY RESULTS FOR THE GRAY-SCALE ALPHA MODELS

Here, we present further visual comparison between the different ALPHA models for gray-scale images, to illustrate the trade-off between quality and diversity.

## E    RELATION BETWEEN ALPHA$_L$ AND ALPHA$_I$

Here, we informally explain why, under some conditions on the wavelet family, setting $\alpha' \in \{0\}$ in the ALPHA models should not lose too much (but still some) information captured by the statistics.

First, remark that the simple linear relation

$$\big(\sum_{\alpha \in \mathcal{A}_4} \rho_\alpha(z)e^{ik\alpha}\big)\rho_0(z) = \sum_{\alpha \in \mathcal{A}_4} \rho_\alpha(z)\rho_0(z)e^{ik\alpha} \tag{9}$$

tells us that computing all correlations for $\alpha \in \mathcal{A}_4$ and $\alpha' = 0$ gives us at least the information contained in the r.h.s. of eq. (9).

Furthermore, if $\alpha \in \mathcal{A}_4 = \{0, \cdots, \frac{3\pi}{4}\}$, we make the following approximation

$$\sum_{\alpha \in \mathcal{A}_4} \rho_\alpha(z)e^{ik\alpha} \simeq \int_{[0,2\pi]} \rho_\alpha(z)e^{-ik\alpha}d\alpha = 2\pi c_k[z]^k.$$

Recall also from the proof of Proposition 2, that $\mathcal{F}(\rho_\alpha(z))(k) = [z]^k c_k$, where the Fourier transform is taken along the variable $\alpha$. Therefore,

$$\rho_0(z) = \sum_{k \in \mathbb{Z}} c_k[z]^k.$$

Therefore,

$$
\begin{aligned}
\big(\sum_{\alpha \in \mathcal{A}_4} \rho_\alpha(z)e^{ik\alpha}\big)\rho_0(z) &\simeq \big(\int_{[0,2\pi]} \rho_\alpha(z)e^{-ik\alpha}d\alpha\big)\rho_0(z) \\
&= 2\pi c_k[z]^k\big(\sum_{k' \in \mathbb{Z}} c_{k'}[z']^{k'}\big) \\
&= 2\pi c_k \sum_{k' \in \mathbb{Z}} c_{k'}[z]^k[z']^{k'}.
\end{aligned}
$$

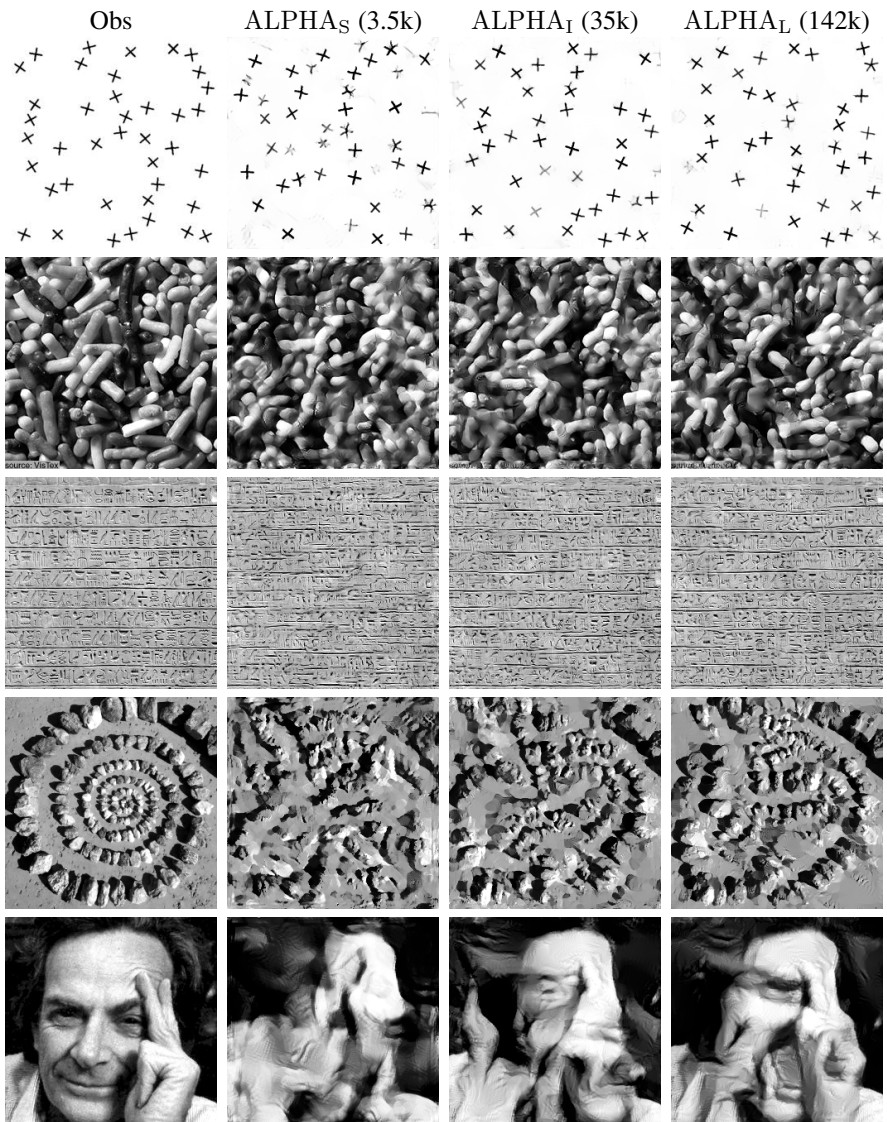

Figure 4: Visual comparison of syntheses from the $\mathrm{ALPHA_S}$, $\mathrm{ALPHA_I}$ and $\mathrm{ALPHA_L}$ models.

Then, replacing $z$ and $z'$ with wavelet coefficients, we get

$$\sum_{\alpha \in \mathcal{A}_A} e^{-ik\alpha} C^{\mathrm{ALPHA}} x((j, \theta, \alpha), (j', \theta', 0), \tau) \simeq 2\pi c_k \sum_{k' \in \mathbb{Z}} c_{k'} C^{\mathrm{WPH}} x((j, \theta, k), (j', \theta', k'), \tau).$$

Using Plancherel's theorem, we can write that

$$C^{\mathrm{WPH}} x((j, \theta, k), (j', \theta', k'), \tau) = \frac{1}{|\Omega_N|} \sum_{\omega \in \frac{2\pi}{N} \Omega_N} \mathcal{F}([x \star \psi_\lambda]^k)(\omega) \mathcal{F}([x \star t_\tau \psi_\lambda]^{-k'})(\omega),$$

where the Fourier transform of an image $x$ is defined by $\mathcal{F}(x)(\omega) := \sum_{u \in \Omega_N} x(u) e^{-i\omega u}$, and $t_\tau$ denotes the translation by $\tau$, i.e. $t_\tau f(\cdot) = f(\cdot - \tau)$.

Now, suppose that the wavelets $\psi_\alpha$ have disjoint compact frequency support, in balls $B_\lambda(2^{-j} C')$, where $\lambda = 2^{-j} r_{-\theta} \xi_0$, and $\xi_0$ is the central frequency of the mother wavelet $\psi$ (cf. ?). Suppose also

that frequency transposition property of the phase harmonics operator (cf. Mallat et al. (2020)) is such that $[x \star \psi_\lambda]^k$ has (approximately) frequency support in $B_{k\lambda}(k2^{-j}C')$. Then, for all $\lambda, \lambda'$, and all $k \in \mathbb{Z}$, there exists only one $k^*$ such that the frequency supports of $[x \star \psi_\lambda]^k$ and $[x \star \psi_{\lambda'}]^{-k^*}$ are not disjoint, i.e. such that $C^{\mathrm{WPH}}x((j, \theta, k), (j', \theta', k'), \tau) \neq 0$. This tells us that

$$\sum_{\alpha \in \mathcal{A}_A} e^{-ik\alpha} C^{\mathrm{ALPHA}}x((j, \theta, \alpha), (j', \theta', 0), \tau) \simeq c_k c_{k^*} C^{\mathrm{WPH}}x((j, \theta, k), (j', \theta', k^*), \tau).$$

Thus, computing all correlations for $\alpha \in \mathcal{A}_4$, and $\alpha' = 0$ gives us (approximately) all the information contained in WPH coefficients for any pair $k, k'$.

This result lies on several approximations, and strong assumptions about the wavelets, which are not fully met in practice. For this reason, setting $\alpha' = 0$ instead of $\alpha' \in \mathcal{A}_4$ effectively reduced the amount of information captured by the statistics, and therefore increases the diversity of the model. However, as we observe in Section 3.2, there is not too much information lost, and the resulting model still captures most of the important geometric structures in texture images.

## F    REDUCED COLOR MODEL

One can reduce the number of statistics in the color model by selection the spatial shift parameter $\tau$ to be non-zero only for correlations between the same color channels. More precisely, it is defined by the following index set: $\Upsilon := \{(\gamma, \gamma', \tau) : ((j, \theta, \alpha), (j', \theta', \alpha'), \tau) \in \Upsilon^{\mathrm{ALPHA_I}}, c = c' \in \{1, 2, 3\}\} \cup \{(\gamma, \gamma', 0) : ((j, \theta, \alpha), (j', \theta', \alpha'), 0) \in \Upsilon^{\mathrm{ALPHA_I}}, (c, c') \in \{1, 2, 3\}^2\}$. This gives a model of size $\sim 113$k, with little degradation of the visual quality, as shown in Figure 5.

## G    VGG SCORE

In Ustyuzhaninov et al. (2017), the authors proposed to use the synthesis loss of the VGG model to evaluate the quality of syntheses from any model. The goal is to define a quantitative, and more objective evaluation method than mere visual inspection. Since the VGG model produces syntheses almost indistinguishable from real textures, it is natural to consider its loss to asses the quality of a synthesis. We computed this loss for the first two examples of Figure 3 (radishes and cherries), and the frist two examples of Figure 9 (gravel and Turbulence flow). Note however that this loss is not exactly the same as the one used in Ustyuzhaninov et al. (2017), as the layers selected to compute the loss are different. In this work, we chose to use the layers suggested in Gatys et al. (2015), (i.e. 'conv1_1', 'pool1', 'pool2', 'pool3', and 'pool4' of the VGG-19 network (Simonyan & Zisserman, 2014)), and compute the relative VGG loss[18].

We notice that this score is not always consistent with visual inspection, as there are texture examples and models for which the syntheses do not look much like the observation image, yet produce a small VGG loss (see e.g. the first and last rows of Figure 3, the RF model syntheses have the smallest loss). It should also be noted that the VGG loss reported on the VGG syntheses is *not* the synthesis loss after optimization, as a histogram matching (HM) procedure is performed as post-processing after optimization. We observed that the VGG loss of the syntheses from the VGG model after HM was considerably higher than the one for syntheses before it, while being visually very similar as illustrated in Figure 6. These observations suggest that the VGG score suffers from instabilities after reaching a certain level (that is, if the VGG loss is small enough, small perturbations of the values of the image pixels might have a strong impact on the loss).

## H    INFLUENCE OF THE CHOICE OF THE WAVELET TRANSFORM

### H.1    INFLUENCE OF THE WAVELET FAMILY

In Section 3.2, we illustrated the importance of the set of indices $\Upsilon$ that define the wavelet coefficients being correlated. Another important role is played by the choice of the wavelets used in

---

[18]Using the code from `https://github.com/ivust/random-texture-synthesis/blob/master/vgg_loss.py` (function style_loss_relative).

Obs                ALPHA$_C$ (320k)        Reduced (113k)

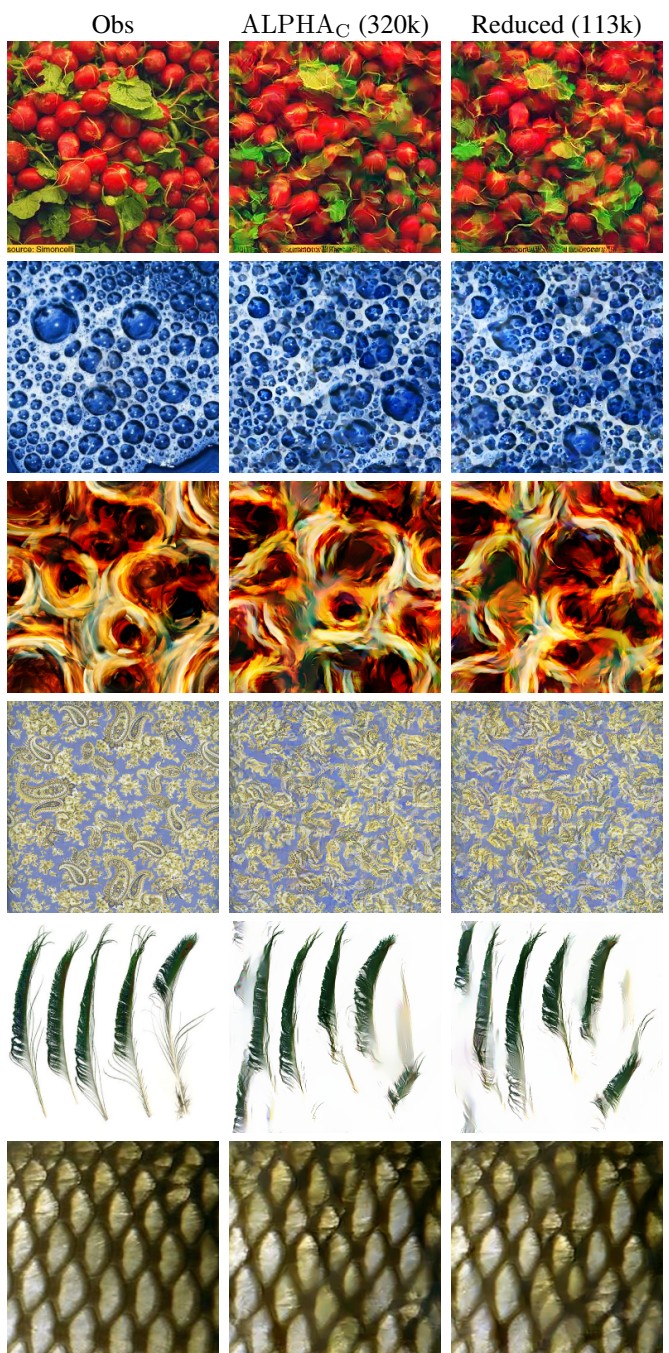

Figure 5: Visual comparison of syntheses from ALPHA$_C$ model and its reduced version.

Table 2: Relative VGG loss of gray-scale textures for the first two examples of Figure 3, and the first two examples of Figure 9.

| Data / Model | ALPHA$_I$ | VGG | PS | RF |
|---|---|---|---|---|
| Radishes | 5.02e-05 | 1.87e-05 | 2.37e-04 | 1.13e-05 |
| Cherries | 4.86e-05 | 1.47e-06 | 6.68e-04 | 9.65e-06 |
| Gravel | 5.97e-05 | 3.08e-06 | 7.25e-04 | 1.29e-05 |
| Turbulence | 5.59e-05 | 5.97e-05 | 2.42e-04 | 3.95e-05 |

Before HM   After HM

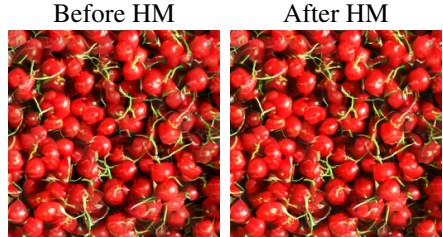

Figure 6: Visual comparison of syntheses from the VGG model, before and after histogram matching. Before HM, the relative VGG loss is 2.22e-08, while after HM, the loss is 5.38e-05.

equation 2. As illustrated in Figure 7, this choice can have a visible impact on the quality of the textures. We observe that, while on the first example, the coherence of the structures appear similar for the three wavelet families, the second example shows that the wavelets used in Portilla & Simoncelli (2000) are less efficient in reproducing the contours of the objects (pebbles). While in our experiments, we chose to use the classical Morlet wavelets, an optimal choice for the wavelet family remains an open problem.

Observation  Simoncelli  Bump  Morlet

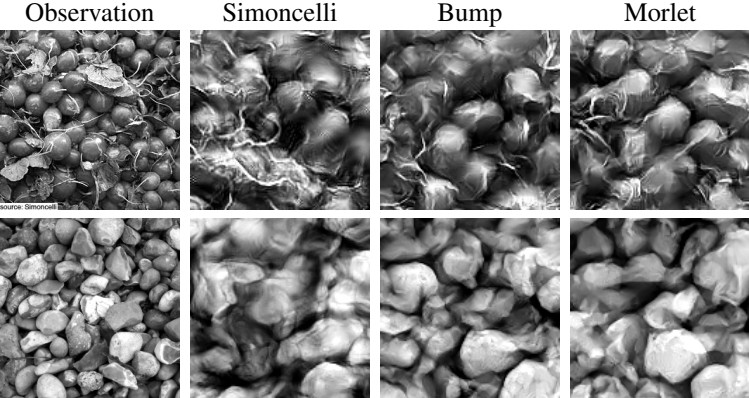

Figure 7: Comparison between different wavelets families used in the $\mathrm{ALPHA}_I$ model. Central zooms of syntheses using the same covariance model, with three different wavelet families. From left to right: observation, Simoncelli steerable wavelets, bump steerable wavelets (Mallat et al., 2020), and Morlet wavelets.

## H.2   INFLUENCE OF SCALE PARAMETER

Recall from Section 2.2.1, the wavelet transform of an image $x$ is defined by

$$\{x \star \psi_{j,\theta},\ x \star \phi_J\}_{0 \leq j < J,\, \theta \in \frac{\pi}{L}\{0,\cdots,L-1\}}.$$

The maximal scale parameter $J$ also plays an important role in the definition of the wavelet transform. It determines the scales of the structures being captured by the transform. If this parameter is too small, large structures in the observation image might not be captured and reproduced in the model syntheses. Conversely, if $J$ is too large, then the large scale statistics may have a high variance, inducing a memorization effect in the syntheses. Figure 8 illustrates this point on two examples from Section 4.2. By setting $J = 4$ (i.e. the maximal range of structures captured by the wavelets is of size $2^4 = 16$), we observe on the first example that the larger structures (bubbles) are not well reproduced. When $J$ is set to 6, the observation is almost identically reproduced by the synthesis. Similarly on the second examples, several parts of the synthesis appear very similar to ones in the observation. We found that a suitable trade-off consists in setting $J = 5$ for images of size $N = 256$.

| Observation | $J = 4$ | $J = 5$ | $J = 6$ |
| --- | --- | --- | --- |

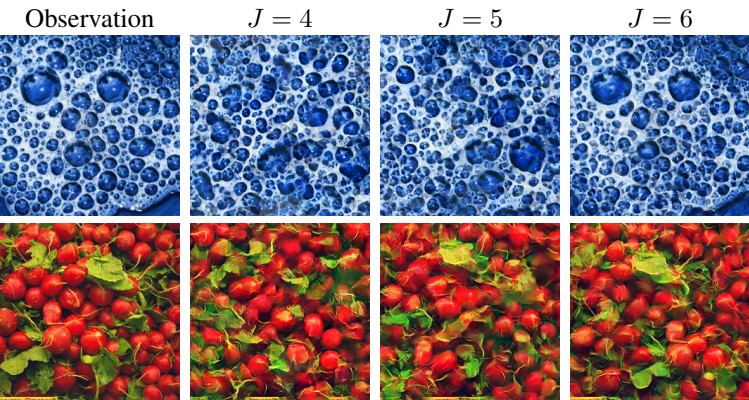

Figure 8: Syntheses from the $\text{ALPHA}_\text{C}$ model defined with three different maximal scale parameters $J \in \{4, 5, 6\}$. As in Section 4, Morlet wavelets are used.

# I  SUPPLEMENTARY RESULTS OF $\text{ALPHA}_\text{I}$/$\text{ALPHA}_\text{C}$ VS. PS, RF, VGG

In Figure 9 we present additional syntheses on various examples, from the PS, $\text{ALPHA}_\text{C}$, RF and VGG models. These examples can be viewed as random (Turbulence flow, tree bark, porous stone), structured (gravel, paisley pattern, tree leaf, school text), or inhomogeneous (crafted pattern of third row, but also the porous stone).

We see that $\text{ALPHA}_I$ again significantly improves the visual quality of the PS model on gray-scale textures such as the gravel and turbulence flow. The visual quality of RF and VGG seems also worse on Turbulence flow compared to $\text{ALPHA}_I$. In some examples such as tree leaf, we find the synthesis of all the models are similar. On the inhomogeneous porous stone, non of the models give satisfying visual results.

Similarly, in Figure 10 are presented supplementary syntheses form the color models, for structured images (radishes, bubbles, flowers), quasi-periodic images (scales, honeycomb, bricks), and non-stationary images (feathers). As previously observed, for highly structured quasi-periodic images such as the bricks example, the VGG model fails to capture long-range correlation, which can be solved using the method of Berger & Memisevic (2017). Syntheses of non-stationary images exhibit memorization effects, as previously observed.

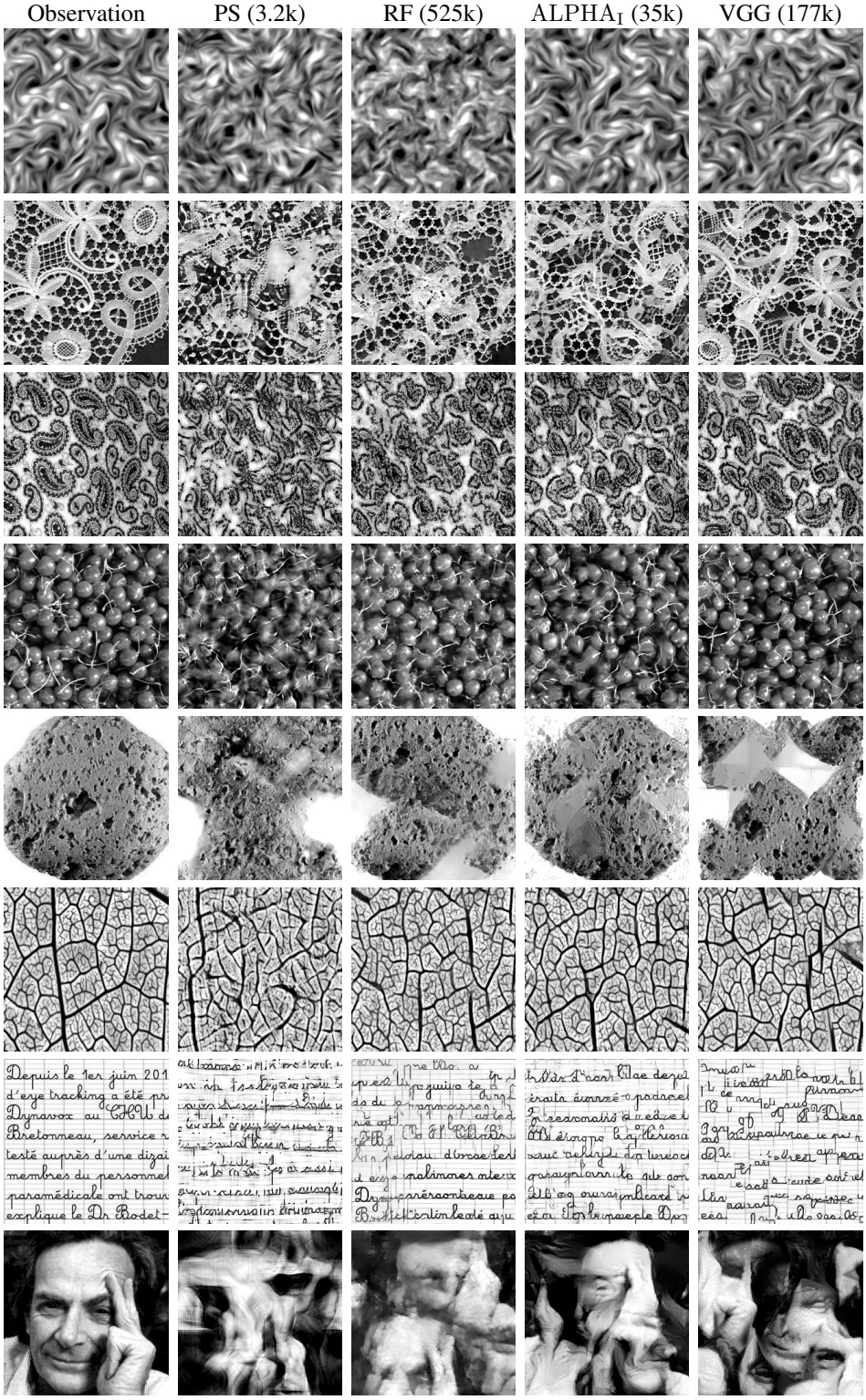

Figure 9: Visual comparison between different texture models on gray-scale images.

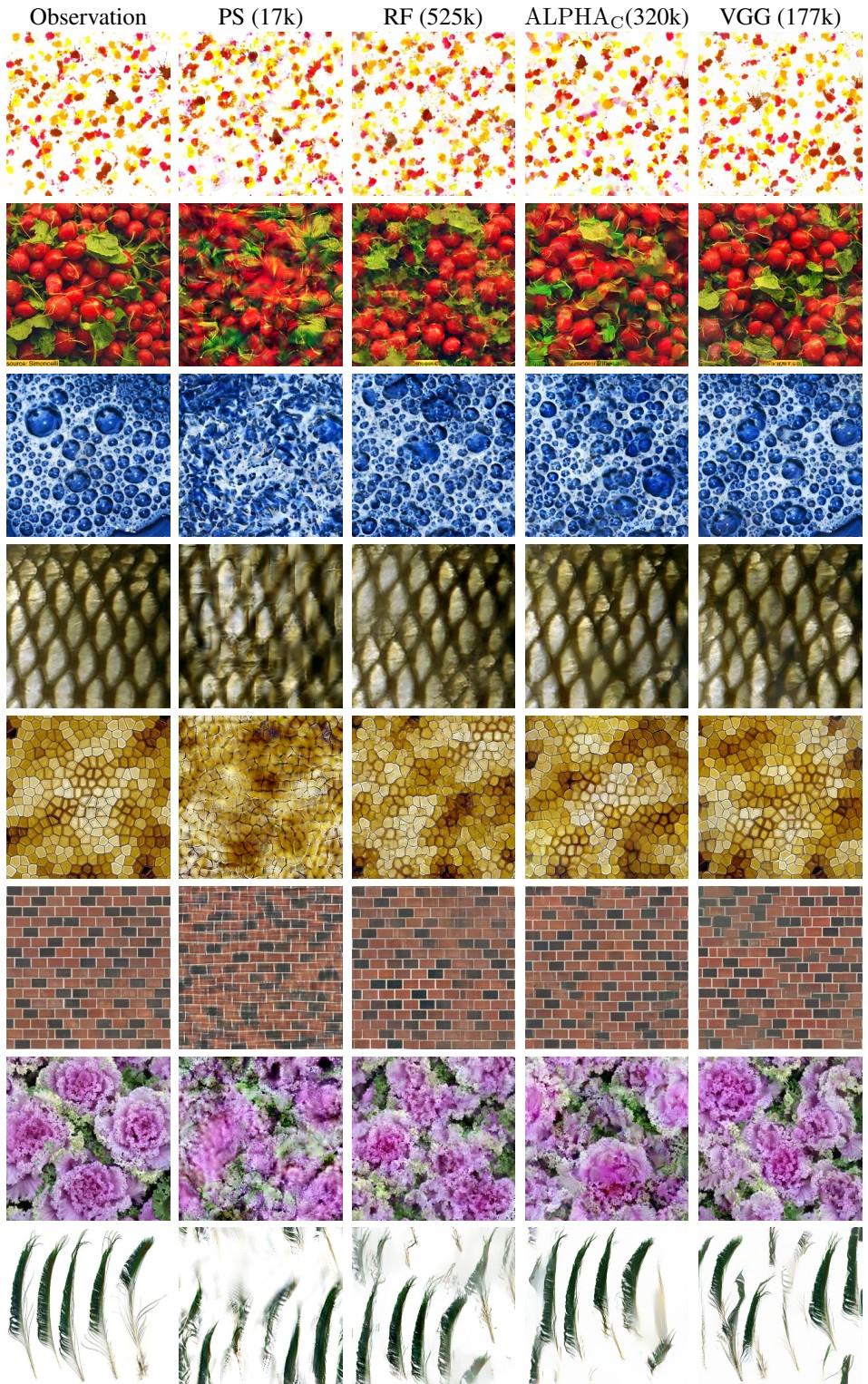

Figure 10: Visual comparison between different texture models on color images.

