# OpenReview forum: "Generalized rectifier wavelet covariance models for texture synthesis"
_ICLR.cc/2022/Conference — ICLR 2022 Poster_

### Official Review · Reviewer_Q8m3 · 2021-11-02

**Correctness:** 4
**Technical Novelty And Significance:** 3
**Empirical Novelty And Significance:** 3
**Recommendation:** 8
**Confidence:** 5

**Main Review:**

Proving the relationship between the different sets of texture models, with the additional proof that ties ReLU's and phase harmonics together is nice work and a strong contribution.  Unfortunately, much of the rest of the paper is lacking and detracts from the overall effect.

Main criticisms:
1.  This paper is not suitable for ICLR as nothing within the paper is learned from data, these are hand constructed (and well though through) representations that are similar to those found in learned representations but are not here learned.  This is not a problem per se, it simply doesn't fit with the conference themes and as such may not be found useful by most of the conference attendees.

2. The paper spends an inordinate amount of space laying out and explaining concepts and algorithms which are not novel contributions of this paper and would be better served with passing references for the reader to follow up with.  The novel contributions of the paper do not begin until the bottom of page 4.

3. The examples shown reveal that all of the Alpha models (including AlphaI and AlphaC the chosen models for analysis in this paper) are memorizing patches of the image (any image that has the portilla simoncelli badge in the bottom left corner has this badge completely recreated somewhere else in the image as completely readable text), which a proper texture model would not do unless it was simply memorizing the image patches.  This is especially apparent in the color images which have more of examples of the name badge in the corner.

4. The authors state parameterizations of comparison models on page 8 that are at least partially incorrect (here stated that portilla simoncelli has ~7000 parameters when in reality it has 701 parameters, an order of magnitude difference).  A careful inspection of the others may find similar magnitude errors.


**Summary Of The Paper:**

The paper presents a novel image texture model that is a hybrid of the traditional wavelet constructions, and the more modern machine learning constructions (incorporating only ReLU nonlinearities on the wavelet basis), proves that this class of models is actually akin to phase harmonic functions and that these are sufficient for capturing texture statistics, and claims to show competitive performance with state of the art with significantly fewer parameters.

**Summary Of The Review:**

For these reasons above I am inclined to reject this paper for ICLR, and suggest the authors find a more suitable conference for this work, as well as reanalyzing some of the results and rethinking the overall presentation of the work.

---

> ### Author Response · Authors · 2021-11-22
> **Response to Reviewer Q8m3**
>
> We thank the reviewer for their time and questions. Below, we provide point by point answers.
>
>
> 1. We acknowledge that the models presented in this work do not rely on any learning mechanism using data, conversely to the model presented in Gatys et al. (2015), that is based on a CNN that has been trained on a classification task on the ImageNet dataset. However, in this work, we compare our models to this learned representation, and discuss the necessity of the learning procedure. Our approach is similar to the work of Ustyuzhaninov et al. (2017), and an extension of the VGG model (Berger and Memisevic, 2017). Both of these works have been published at ICLR. Moreover, as suggested by another reviewer, we find this topic is connected with other papers that are published in machine learning conferences such as Neurips. Therefore we sincerely believe that ICLR conference is a coherent venue for the subject of our work.
> 2. The framework for statistics-based texture synthesis, and the mathematical background on the construction of wavelet-based representations was indeed too long, and our contributions should be presented to the reader sooner in the paper. Therefore, we considerably shortened the Section 2.1, and moved some details about the WPH and PS models (of the original Section 2.2.2) to Appendix A.
> 3. For observation textures containing a logo (image badge) at the bottom left corner, this logo is indeed reproduced in our syntheses. However, this is most likely due to the handling of borders in the model (we put zero values to pixels at the border of non-periodic images), rather than to a genuine memorization effect. We think such reproduction of certain border patches does not significantly affect the diversity of the model, which is now further discussed in the paper (Section 4.2). This phenomenon is also present in the state-of-the-art models (RF and VGG) which use a similar handling of borders. For textures that present stationary structures at the border (without logo), non memorization at the border is visible.
> 4. The number of statistics in the PS model depends on a certain number of parameters, such as the number of scales, orientations, or the size of the spatial shift. In our experiments, the number of statistics is not 701 because of our choice of parameters. We have used a larger set of statistics, as it provides syntheses with better visual quality. However, due to the redundancies in the covariance statistics of the PS and RF models, some statistics had been counted twice (however, the order of magnitude remains the same). These numbers are now revised, and the more details on the computation of the number of statistics of the different models is now given in Appendix A.

---

> > ### Comment · Reviewer_Q8m3 · 2021-11-26
> > **Thanks to the authors**
> >
> > I'd like the thank the authors for their revisions and clarifications.  I agree that the memorization effect observed is likely due to the boundary handling and am satisfied by the modification to the text.  I also appreciate the additional figures added (at the request of another reviewer), they provide more insight that adds to the value of this paper.
> >
> > Thank you also for spending more space of the paper highlighting the direct contributions of this work, the paper is stronger for it.
> >
> > I will raise my review because I do believe this work is quite good, but my reservations about audience fit still hold.  In my opinion, this would be a very strong NeurIPS paper, which has more space for non-learning based work on neural representations.  But your point stands, similar work has been published at ICLR in the past and this work does compare to learned representations.

---

### Official Review · Reviewer_Gwxo · 2021-11-02

**Correctness:** 4
**Technical Novelty And Significance:** 3
**Empirical Novelty And Significance:** 3
**Recommendation:** 8
**Confidence:** 3

**Main Review:**

**Positive points**: The results presented in the paper are very interesting because they shed light on what exactly are CNNs learning on images, since the presented model can produce similar results to state of the art CNNs with a much smaller number of parameters and a fixed filter structure (wavelets). This is, obviously very important given the usage of CNNs in the literature and the little current comprehension there is on the information they encode and it is relevant for image comprehension. In this sense, this paper shows that the rectifier wavelet operator is

**Negative points**: The relationship with patch-based methods is not discussed at all. Even thought the proposed algorithm lays within the "statistics-based methods", in the numerical results it should be compared to other family of methods such as exemplary-based models (see Raad  et al 2018 for examples)

**Summary Of The Paper:**

This paper presents a new image representation model based on wavelets and non-linear rectifiers that allows to synthesize complex geometric textures with a better visual quality than previous wavelet-based models.
The main interest of the paper is the usage of the mathematical results from Mallat et al 2020 and Zhang and Mallat 2021 on wavelet phase harmonics to image texture synthesis.

They also show that the PS model (Portilla & Simoncelli 2000) is a particular case of the wavelet phase harmonics-based (WPH) model. Both methods underline the importance of statistical dependencies between wavelet coefficients across scales. The specific choice of the parameters available for WPH is important to balance between reproducing good structural information and memorizing patterns from the observation.


**Summary Of The Review:**

The paper presents the results on texture synthesis using a new image representation model based on wavelet phase harmonics. The results are interesting and show that these representations are powerful enough to encode complex image features. What this paper presents starts clarifying how CNNs are encoding image comprehension

---

> ### Author Response · Authors · 2021-11-22
> **Response to Reviewer Gwxo**
>
> We thank the reviewer for the positive feedback. The relation between statistics-based approaches (e.g. our maximum-entropy models) and patch-based approaches is indeed a very interesting question.  We have thus mentioned it in the introduction by citing a recent work which uses the type of statistics that we study in the paper to improve patch-based methods. Therefore these two approaches are complementary. We believe that the comparison between the two methods constitutes a whole work in itself, as they rely on quite different approaches that lead to comparable results.

---

### Official Review · Reviewer_bssY · 2021-11-02

**Correctness:** 4
**Technical Novelty And Significance:** 3
**Empirical Novelty And Significance:** 3
**Recommendation:** 8
**Confidence:** 5

**Main Review:**

This is a paper that has a great idea going on with regards to qualitatively estimating the minimum number of constraints in a parametric texture model that revolves around wavelets. My main motivation for marginally rejecting this paper is that it comes up short in terms of the types of qualitative insights it can deliver, not capitalizing enough in the rich mathematical motivation for a wavelet-constrained model. Nevertheless, I think I am lenient on increasing my score if authors address some of the points I mention below and I’m also genuinely curious on what the other reviewers thoughts are.

* Not enough highlight on the number of parameters of the wavelet based model. This should be emphasized more in Table 1, or in an extra table as it’s superficially mentioned at the end of section 4.1. The exploration of how the number of parameters affects the quality of the synthesized stimuli is a pivotal part of this paper that has not been highlighted enough (the results are there, but the presentation needs to improve).
* Adding more examples of model failures, and singular successes. The figures in 2,3, and 4 are too basic and standard. I think they should be resumed in another separate figure  (perhaps just using figure 4 or a variation of this), and there should be *other new* figures that highlight how the models perform for very specific images. What would the synthesized textures look like for a scene? An abject? Julesz conjecture counter-examples? And does the quality change as the number of parameters is varied (Alpha S,I,L,C) Many of the previously mentioned images that do not provide a stationary signal can magnify the differences in terms of how each model computes its perceptual constraints.
* Linking to the previous bullet point: Why does each model fail? What texture constraint has varied, such that the model all of a sudden breaks down or succeeds for some of these other images that are different from the canonical textures shown in the texture synthesis literature.

Overall, the paper is well written, it provides a good mathematical overview of wavelets and also cites the relevant literature on this subject. I would also suggest authors to look into citing this paper as well that provides some connections of texture synthesis to neuroscience:

* Texture Interpolation for Probing Visual Perception. Vacher et al. NeurIPS 2020.

Once again, I am looking forward to reading the author's rebuttal as I am eager to increase my score as this paper is already heading in the right direction.


**Post-Rebuttal Notes**: I wanted to increase my score from 5 to 7, but since there is no 7 option I have decided to sin on the generous side and give this paper an 8. [I apologize to authors and/or reviewers if they have seen me toggle back and forth between 6 and 8. My overall vote is that this paper be accepted as a poster.]



**Summary Of The Paper:**

This paper provides an attempt to characterize wavelet-based texture synthesis models contingent on the number of parameters and the nature of the parametrization. Proper baselines are added with respect to other relevant texture synthesis models.

**Summary Of The Review:**

A solid first step, but more qualitative insights are needed given to make this paper shine as much as it should. Weak Reject with a possibility of increasing my score during rebuttal to accept.

---

> ### Author Response · Authors · 2021-11-22
> **Response to Reviewer bssY**
>
> We thank the reviewer for the helpful and insightful feedback. We have taken into considerations all the suggestions, and we believe the current revised version has significantly improved. Below, we provide point by point answers to the questions.
>
>
> 1. The number of statistics in each wavelet model is indeed a central aspect of these representations. We have now highlighted this aspect in the manuscript, by adding in Table 1 the number of statistics as a function of the parameters for the different ALPHA models. For more clarity, we report the effective number of each model next to its name at the beginning of each figure. Additionally, to illustrate the fact that the choice and number of statistics is a crucial matter and an open problem, we now present a reduced version of the color model, containing about three times less statistics, giving similar visual quality to the ALPHA_C model. The details are given in Appendix  F.
> 2. To better understand the role of the different statistics in the ALPHA models, we added new examples in Figure 2, showing how the models behave on a broad range of images, such as a classical Julesz conjecture counterexample, and a non-stationary image exhibiting strong correlations across scales and orientations. Furthermore, Figure 3 and 4 have been regrouped, and we added new examples of non-stationary images (such as scene or object), illustrating how the different models build their perceptual constraints, and providing insights to subtle memorization effects. Additional examples have also been added in the supplementary material (Appendix D and I).
> 3. To provide more insights, Figure 2 illustrates the important impact of the correlations between spatially shifted coefficients across different scales and orientations to capture the geometry of salient structures and objects in images. It gives a better intuition about why the small model ALPHA_S fails to reproduce several geometric aspects of texture images. Furthermore, we added a new synthesis example showing that the larger model ALPHA_L can suffer from memorization effects. A mathematical motivation for going from the large model to the intermediate model by setting the phase parameter alpha’=0 is now added in the manuscript, and is detailed in Appendix E.
> 4. The paper Vacher et al., NeurIPS 2020 suggested by the reviewer gives a broader perspective of the paper. We have now added it to the introduction of the paper.

---

> > ### Comment · Reviewer_bssY · 2021-11-24
> > **Updating Score**
> >
> > Dear Authors,
> >
> > Thank you for your reply! I am quite happy with how the paper has turned out as it looks like it is in better shape than the initial submission. I specially appreciate the updated figures with non-stationary images. I've decided to increase my score from 5 to 8. I think this paper would make a good poster at ICLR as it clearly has some ups/downs partitioned between the reviewers! I think the partitions is split because half the reviewers/audience may come from a computer vision background while the other from a neuroscience background (though this is not a bad thing!)
> >
> > What I think can and should be pushed further is: **Why should we even care about having good texture models for stationary and non-stationary images in the first place?** Similar to work of Vacher et al, ongoing works of Visual Metamerism would greatly benefit from these types of models, and I'd further suggest citing:
> >
> > * Freeman & Simoncelli (Metamers of the Ventral Stream. Nature Neuroscience, 2011)
> > * Deza et al. (Towards Metamerism via Foveated Style Transfer, ICLR 2019)
> > * Wallis et al. (Image Content is more important than Bouma's Law for Scene Metamers. eLife 2019)
> >
> > Works like the ones above should be added in the Introduction to further motivate the impact of this paper.

---

> > > ### Author Response · Authors · 2021-11-29
> > > **Rebuttal answer to Reviewer bssY**
> > >
> > > We thank the reviewer for their positive feedback. We will take into account their citation suggestions in the paper.

---

### Official Review · Reviewer_xfp4 · 2021-11-20

**Correctness:** 3
**Technical Novelty And Significance:** 2
**Empirical Novelty And Significance:** 3
**Recommendation:** 3
**Confidence:** 4

**Main Review:**

This paper provides a detailed review and explanation of wavelet and how to use the rectified wavelet coefficients for covariance statistics matching based texture synthesis. The motivation also seems to be clear: trying to use less number of statistics to achieve satisfying quality. However, I have several main concerns regarding to the comparisons with recent papers and the types of texture images this method can handle.

1. as the main motivation is trying to achieve good quality with as few number of statistics as possible, from figure 3 and figure 4, the quality of “VGG” seems to be better than the proposed method and also uses less number of statistics (much less for color textures with 177k vs. 320 and similar for gray texture 177k vs. 142k)

2. After Gatys et al. 2015 and Ustyuzhaninov et al. 2017, there are more recent methods on texture synthesis which have achieved many impressive results. For example:
    Deep correlations for texture synthesis.  Sendik & Cohen-or et al. 2017.
    Texture Synthesis Through Convolutional Neural Networks and Spectrum Constraints. 2016.
    Non-Stationary Texture Synthesis by Adversarial Expansion. Zhou et al. 2018.
    and many others.
  This submission did not cite or compare with these approaches (and other more recent methods).

3. The submission and supplementary file provide very limited number of results. The images are mostly stationary images. More examples of structure images and non-stationary images are needed to validate the proposed method.

4. One main application of texture synthesis to generate a larger image output given a smaller input patch. It seems that the proposed can not generate a larger image output.


**Summary Of The Paper:**

This work proposed a texture synthesis framework using the rectified wavelet coefficients. The paper claims that the proposed method cand achieve similar quality with the VGG feature based method (Gatys et al. 22015) and random filter based method (RF, Ustyuzhaninov et al 2017) and gets better quality than PS (Portilla & Simoncelli 2000, while requiring less number of statistics than RF.

**Summary Of The Review:**

Although the motivation of the proposed framework seems to be pretty specify and properly positioned, the experiments does not seem to validate the superiority over existing methods. The results are quite limited. It also missed the citation and comparisons with many recent works.

---

> ### Author Response · Authors · 2021-11-22
> **Response to Reviewer xfp4**
>
> We thank the reviewer for their time. We hope that, despite the late arrival of the reviewers questions, our answers will prove satisfactory.
>
> We believe that the reviewer may have misinterpreted the main objective of this work . We have now modified the introduction to make this point clearer. Our goal is not to achieve similar results than state-of-the art models with fewer statistics, but rather to build a wavelet-based model, hence with more interpretable statistics than CNN-based models (VGG and RF). We aim at bridging the gap between the classical work of Portilla and Simoncelli (2000), and these CNN-based state-of-the art models. This is motivated by the open problem (Juelsz conjecture) of finding what statistics are needed to describe the geometric structures in natural textures, which relates to the number of statistics of the model. The number of statistics is important in our model, as we illustrate the trade-off that too few statistics may fail to capture important information, and too many may result in a lack of diversity.
>
> Below, we provide point by point answers to the reviewers questions.
>
>
> 1. We have now made the number of statistics of each model easier to read in the paper. Our proposed model for gray-scale images has ~35k statistics (the ALPHA_I model), not 142k, which corresponds to the larger model for which we observe memorization issues. Thus our gray-scale model has a much smaller size than the VGG model. The color model has indeed a larger number of parameters than VGG, however we believe that they are in the same order, and there are possibilities to reduce this number, as we incorporated all correlations between color channels. To substantiate this, we have now introduced a reduced color model that has ~113k statistics, that results in syntheses of similar visual quality. We believe that further reduction of the number of statistics is still possible.
> 2. In this paper, we do not tend to cite too many papers, as we have already cited a recent review paper (Raad et al., 2018) on the topic.  The papers suggested by the reviewer indeed provide impressive results relatively to the VGG model, but our goal is different. We have now cited them in the conclusion as we find that they are complementary to our work. The VGG and RF models are the most relevant baselines in our opinion, as they are based on the same framework of maximum entropy models built on covariance statistics.
> 3. We have now added additional syntheses, both in the main manuscript and in the supplementary material. In particular, we show more examples of syntheses with structures, and also syntheses of non-stationary images.
> 4. Synthesis of larger textures is indeed an interesting problem, but it is out of the scope of this paper. We think it is possible to synthesize larger textures, by initializing the synthesis with a larger input, and using the statistics computed on the smaller observation. Additionally, our model is closely related to the PS model, with which such applications are possible (see Fig 20 in Raad et al., 2018), therefore such standard applications should also be possible.

---

> > ### Comment · Reviewer_xfp4 · 2021-11-27
> > **Keep the rating**
> >
> > I would like to thank authors for the efforts in the revision and rebuttals, even though the review was submitted later because the review invitation was received after the reviewing deadline (several days before the review was submitted). I also appreciate the author’s statement of proposing a more interpretable representation using wavelet-based model for texture synthesis. However, I still feel skeptical about the submission due to the following reasons:
> >
> > 1. Citing a review paper published 3 years ago should not be the reason to ignore the comparisons with more recent works, especially “Deep correlations for texture synthesis. Sendik & Cohen-or et al. 2017.”, which presented a better representation than the VGG-based gram matrix. There are also many other works like: “Texture Mixer: A Network for Controllable Synthesis and Interpolation of Texture. Yu at al. 2019”, which proposes new numerical metrics to evaluate textures synthesis quality.
> >
> > 2. In “Deep correlations for texture synthesis. Sendik & Cohen-or et al. 2017”, the author proposed to use Deep Correlation Map to guide the texture synthesis, which has clearly surpassed the VGG based method, especially in preserving the structures in the output. The deep correlation map is closely related to VGG and has shown superior output quality than VGG-based gram matrix. Thus, the author only compares with random filters and VGG should not be enough, since these are not the state-of-the -art methods as claimed in the submission.
> >
> > 3. Qualitative Comparison: for most of the results shown in the submission, I did not see the outputs from the proposed method would not (clearly) outperform VGG visually. The outputs from Deep Correlation Map are also supposed to be much better from the textures with some structure patterns. For example, for Figure 10’s structured texture image, the output from Deep Correlation Map should be much better.
> >
> > 4. Quantitative Comparison: as the author mentioned in Appendix G, the VGG score is not a good numeric measurement. The synthesis process of “VGG” method is also closely related to the VGG loss(score). In other words, the manual threshold for terminating the optimization iterations will also closely affect the final VGG score. On the other hand, similar to author’s motivation of proposing a more interpretable representation, the work “Texture mixer: A network for controllable synthesis and interpolation of texture” has also proposed some more interpretable numeric metrics. Using these or some similar metrics may lead to more comprehensive and more interpretable quantitative comparisons.
> >
> > For the question about “Why should we even care about having good texture models for stationary and non-stationary images in the first place”, there is no such requirement for each method to perform well on non-stationary texture images. It is just the fact that some recent texture synthesis papers have specifically achieve impressive results on non-stationary texture synthesis, like “Non-stationary texture synthesis by adversarial expansion. Zhou et al.”. It is fine for a new method not to perform well on non-stationary texture images by stating the limitation clearly. Besides the non-stationary texture images, structure/regular texture images are also important and would distinguish the performance of different methods very well.
> >
> > In short, I believe that to propose a new method, it would be necessary to compare against the recent related methods more thoroughly. I would keep my rating.

---

> > > ### Author Response · Authors · 2021-11-29
> > > **Rebuttal answer to Reviewer xfp4**
> > >
> > > We thank the reviewer for their additional comments. We respectfully disagree with the reviewer’s point of view, for the following reasons:
> > >
> > >
> > > 1. As we have stated, we find it more reasonable to cite the work “Deep correlations for texture synthesis. Sendik & Cohen-or et al. 2017.” (and other related works) in the conclusion, because we consider it as a possible extension of our current work. The model in Sendik & Cohen-or et al. 2017 is beyond the micro-canonical framework that we have reviewed in Section 2 of our paper, therefore we find it is beyond the scope of the models that we study. Indeed the synthesis model is based on minimizing a sum of 4 different losses (eq. 12 in the Deep Correlations paper of 2017), min E = alpha E_DCor  + beta E_Grm + eta E_Div + gamma E_Smooth. The term E_Smooth depends only on the synthesis and thus it is no longer matching two set of statistics as in the micro-canonical model. These new losses could also be added to our model, but it is beyond the scope of this paper (our goal is to define mathematical and interpretable statistics that are competitive with CNN based representations).
> > > Regarding the citation “Texture Mixer: A Network for Controllable Synthesis and Interpolation of Texture. Yu at al. 2019”, we find that it is not very relevant to cite since the metrics proposed in Yu at al. 2019 all seem to be about the evaluation of the visual quality of the synthesis (user controllability, interpolation smoothness, and interpolation realism). They do not allow one to measure the diversity of a model, which is a central interest of our paper, as we aim to achieve a balance between quality and diversity.
> > >
> > > 2. We found that the syntheses from the Deep Correlations model only outperforms the VGG model on a certain type of textures, either periodic or with aligned structures. In Figure 9 of their paper (Sendik & Cohen-or et al. 2017), in our opinion, the 3 last examples (rows) are not better synthesized with their model. On the contrary, we find that VGG achieves the best visual quality. This indicates that this model is not overall better than the VGG model, but only in certain cases. Such observation is consistent with the conclusion in the review paper of Raad et al. 2018 (see Section 5.2).
> > >
> > >
> > > 3. We do not claim that our method visually outperforms the VGG model. Even though the VGG model fails to reproduce some long-range or periodic structures in the tiles example of Fig. 10, we have already discussed in the paper that additional constraints (see Berger & Memisevic, 2017) can remedy this issue.
> > >
> > >
> > > 4. As stated in our first point, the metrics proposed in “Texture Mixer: A Network for Controllable Synthesis and Interpolation of Texture. Yu at al. 2019” do not seem to take into account the diversity of the evaluated models. Therefore, in terms of the evaluation, it remains open to find a quantitative evaluation measuring the quality vs. diversity trade-off, as we have mentioned in the conclusion of the paper.
> > >
> > > According to Raad et al. 2018,  “Grenander proposed to use the term “texture" for strictly stationary stochastic processes. Giving a precise definition of textures is a slippery task; in a sense, each model implicitly proposes one …”. In this paper, we consider textures as stationary rather than non-stationary. Our syntheses of non-stationary images only have an illustrative purpose. Therefore, it is out of the scope of this paper to compare our model with specific models for non-stationary cases, such as  “Non-stationary texture synthesis by adversarial expansion. Zhou et al.”, proposed by the reviewer.

---

### Comment · Area_Chair_cghF · 2021-12-05
**Follow-up questions**

To Authors:

As to the related works, can you further clarify the relationship and difference between your model and the following two:

[a] Minimax entropy principle and its application to texture modeling. Neural Computation, 1997

[b] A Theory of Generative ConvNet. ICML 2016

Both of them are maximum entropy model in the form of Markov random field. [a] is based on statistics defined by Gabor wavelets, and [b] is based on statistics defined by CNN response. We find that [a][b] are related to your work but the current paper lack a discussion and comparison with them. [a] is the first maximum entropy model that uses wavelets for modeling texture, while [b] is the first maximum entropy model that uses CNN for modeling texture. Both [a][b] learn intrinsic statistics (trainable coefficients in [a] and trainable CNN in [b]) to represent texture.

---

> ### Author Response · Authors · 2021-12-07
> **Answers to Area Chair cghF**
>
> We thank the Area Chair and Reviewers for their questions. We are aware of these papers, however, as there is a large amount of papers for texture modeling in the literature, we have chosen the most important ones that are related to our work. Nevertheless, we are glad to cite these papers (including the ones suggested by the Reviewer xfp4), by highlighting the main differences as they are quite different from our approach.
>
> The mini-max entropy principle (proposed in [a]) is a central idea to optimize the statistics of a maximum-entropy model. However, the major difference between the methods from [a, b] and ours is that their models are adapted to the texture (or image) that they want to synthesize. In [a], the authors look for a set of statistics adapted for each texture (e.g. by optimizing the wavelet filter bank); in [b], the representation is learned from a CNN, based on a given texture. Conversely, our method, as well as the methods of PS, RF and VGG, seek to find a ‘universal’ set of statistics, that is able to describe all possible geometric structures in texture images. This line of research, based on the Julesz conjecture that there exists a set of statistics describing all visually distinguishable textures, is thus very different to the learning-based approaches of [a, b] and other more recent ones (e.g. Cooperative Training of Descriptor and Generator Networks, TPAMI, 2018). According to Portilla and Simoncelli (2000), the learning-based approaches are based on a more restricted vision of the Julesz conjecture.
>
> Additionally, the FRAME model developed in [a] uses simple wavelet-based statistics (e.g. marginal histograms). The use of wavelet-based statistics was a hot topic at the time (see e.g. Computational Modeling of Visual Texture Segregation (1991)). In Portilla and Simoncelli (2000), the authors advocate that the use of joint correlations between wavelet coefficients is more adapted to the synthesis of a broad range of textures, and show that it gives better visual quality, see e.g. Figs 7 and 8 in [a], compared to Figs 12 and 14 in Portilla and Simoncelli (2000). Such correlations statistics are also used in the works of Gatys et al. (2015) and Ustyuzhaninov et al. (2017), and are the basis of our work.
>
> On a minor note, the models from [a] and [b] are macro-canonical models, i.e. they seek to sample from a maximum entropy model with expectation constraints, hence the Gibbs distributions. For this reason, [a] uses proposes a Gibbs sampler or MCMC methods to sample their syntheses, and [b] uses Langevin dynamics. This is different from the micro-canonical setting (on which the models from Portilla and Simoncelli (2000), Gatys et al. (2015), and Ustyuzhaninov et al. (2017) are based on), approximated by gradient descent.
>
> For all these reasons, we believe it is more relevant for us to compare our model to the works of Portilla and Simoncelli (2000), Gatys et al. (2015), and Ustyuzhaninov et al. (2017).

---

### Decision · Program_Chairs · 2022-01-20

**Decision:**

Accept (Poster)

**Comment:**

This paper proposes a new wavelet-based model to represent textures. The model incorporates a wide range of statistics, by computing covariances between rectified wavelets coefficients, at different scales, phases and positions. The model can synthesize textures that have a similar quality to state-of-the-art texture models using CNN structure. Qualitative results are shown to demonstrate the effectiveness of the model.  The paper studies an important problem in computer vision and neuroscience, which is texture modeling. However, many important related works are missing. After rebuttal, three of four reviewers champion accepting the work because the proposed wavelet-based texture model, which produces competitive synthesis with much less parameters than the CNN-based model, will be beneficial to the fields of computer vision and neuroscience. One reviewer has critical comments on this paper because the paper lacks a comparison again more recent works both quantitatively and qualitatively. However, during rebuttal, the authors expressed their disagreement with it and pointed out that the goal of the paper is to bridge the gap between the classical work of Portilla and Simoncelli (2000), and the CNN-based models and to find what statistics are needed to describe the geometric structures in natural textures. Their discussion didn't reach an agreement after rebuttal. After an internal discussion, AC recommends accepting the paper but urges the authors to improve their paper by taking into account all the suggestions from reviewers, especially include the discussion or comparison with those related works mentioned in the rebuttal.